Open Access EGU

# The new implementation of a computationally efficient modeling tool (STOPS v1.5) into CMAQ v5.0.2 and its application for a more accurate prediction of Asian dust

Wonbae Jeon[1], Yunsoo Choi[1*], Peter Percell[1], Amir Hossein Souri[1], Chang-Keun Song[2], Soon-Tae Kim[3], and Jhoon Kim[4]

[1]Department of Earth and Atmospheric Sciences, University of Houston, 312 Science & Research Building 1, Houston, TX 77204, USA

[2]National Institute of Environmental Research, Inchon, Republic of Korea

[3]Division of Environmental Engineering, Ajou University, Suwon, Republic of Korea

[4]Department of Atmosphere Sciences, Yonsei University, Seoul, Republic of Korea

Correspondence to: Yunsoo Choi (ychoi6@uh.edu)

**Abstract.** This study suggests a new modeling framework using a hybrid Lagrangian-Eulerian based modeling tool (the Screening Trajectory Ozone Prediction System, STOPS) for a more accurate prediction of Asian dust event in Korea. The new version of STOPS (v1.5) has been implemented into the Community Multi-scale Air Quality (CMAQ) model version 5.0.2. We apply STOPS to $PM_{10}$ simulations in the East Asia during Asian dust events (22-24 February, 2015). The STOPS modeling system is a moving nest (Lagrangian approach) between the source and the receptor inside a CMAQ structure (Eulerian model). The proposed model generates simulation results that are relatively consistent with those of CMAQ but within a comparatively shorter computational time period. We evaluate the performance of standard CMAQ for the $PM_{10}$ simulations and investigate the impact of STOPS modeling with constrained PM concentration based on space-derived measurement (by using alternative PM emissions) on the improved accuracy of the $PM_{10}$ prediction. We find that standard CMAQ generally underestimates $PM_{10}$ concentrations during the simulation period (February, 2015) and fails to capture $PM_{10}$ peaks during Asian dust events. Accurately simulated meteorology implies that the underestimated $PM_{10}$ concentration is not due to the meteorology but to poorly estimated dust emissions for the CMAQ simulation. To improve the underestimated $PM_{10}$ results from standard CMAQ, we use the STOPS modeling system inside of the CMAQ model, and instead of running the costly, time-consuming Eulerian model, CMAQ, we run several STOPS simulations using constrained PM concentration based on aerosol optical depth (AOD) data from Geostationary Ocean Color Imager

(GOCI), reflecting real-time initial and boundary conditions of dust particles near the Korean Peninsula. The STOPS simulations with constrained PM concentration by GOCI-derived AOD show a significant increase in simulated $PM_{10}$ compared to standard CMAQ. Moreover, the STOPS results were closely matched to surface data. These promising results imply that STOPS could prove to be a useful tool for more accurately predicting Asian dust events in Korea. With additional verification of the capabilities of the methodology on concentration estimations and more STOPS simulations for various time periods, the benefit of STOPS modeling for more accurate predictions of Asian dust could be generalized to the simulation and forecasting of unexpected events such as wildfires and upset emissions events in industrial regions over the East Asia.

## 1 Introduction

One of the major air pollutants in the lower atmosphere is particulate matter (PM). Numerous studies have reported its adverse effects on human health and the environment (Park et al., 2005; Heo et al., 2009; Jeon et al., 2015). Extreme levels of PM and the frequent occurrence of high PM events in the East Asia region have become a major social issue, particularly in South Korea (Korea, hereafter), geographically located in downwind from China and several desert areas, which are the source of enormous quantities of emissions. Severe PM events associated with long-range transport of these emissions that originate primarily in Mongolia and the Gobi Desert (Chun et al., 2001; Kim, 2008; Heo et al., 2009) cause extraordinarily severe yellow sand storms that often cover the entire sky over Korea during the spring and late winter. The pollutants in the Asian dust result in the reduced visibility (Chun et al., 2001) and increased mortality due to cardiovascular and respiratory diseases (Kwon et al., 2002), and their adverse effects become more evident in cities closer to source regions of the Asian dust (Kashima et al., 2016).

In response to the problems resulting from Asia dust, the Ministry of Environment of Korea has undertaken $PM_{2.5}$ and $PM_{10}$ forecasting since 2015 to prevent possible harm caused by high PM concentrations; but the forecasting, however, sometimes fails to capture high-level PM events. Accurate PM forecasting is challenging because of the complicated physical and chemical properties of PM and the numerous factors such as meteorology and emissions that change PM concentrations (Gelencser et al., 2007; Kim et al., 2008; Tie et al., 2009).

A number of studies have described the important role of meteorology in PM simulation (Pai et al., 2000; Otte, 2008a; Otte, 2008b), and some have suggested a variety of optimization techniques for enhancing the accuracy of meteorology (Ngan et al., 2012; Lee et al., 2011b; Choi et al., 2012; Jeon et al., 2014; Jeon et al., 2015; Li et al., 2016). In addition, accurate and updated emission inventories are essential to more accurate PM forecasting. Several studies have used anthropogenic emissions inventories for the Asia domain, such as the International Chemical Transport Experiment - Phase B

(INTEX-B) emissions inventory in 2006 and a mosaic Asian anthropogenic emissions inventory in 2010 (MIX) for reliable model performance (Zhang et al., 2009; Zhao et al., 2012; Li et al., 2015). In reality, the use of the optimized meteorology and the most recent emissions inventory as input data for PM simulations can provide accurate forecasting results for the time periods without any specific or unexpected events (e.g., Asian dust). However, predicting the transport of severe sand storms from source regions during the Asian dust events is difficult because of the high uncertainty of dust emissions. Therefore, accurate calculations of dust emissions is essential for more accurate prediction of Asian dust events, but no standardized emissions inventory of dust is currently available because of its high variability primarily caused by the synoptic and local weather conditions near the desert areas.

To address this issue, the intent of this study is to introduce a modeling tool for PM simulation that can be used with the Community Multi-scale Air Quality (CMAQ) model to more accurately predict PM levels. We will apply a hybrid Eulerian-Lagrangian model, the Screening Trajectory Ozone Prediction System (STOPS): a moving nest domain between the source and the receptor inside a CMAQ structure), to simulate PM in the East Asia region. STOPS provides simulation results similar to those of CMAQ, but it does so much faster than the full CMAQ modeling system. The detail of original version of STOPS (v1.0) and its benefit for regional air quality study was first introduced by Czader et al. (2015). However, since STOPS v1.0 was based on CMAQ v4.4, it can hardly be used for recent PM study due to outdated modules and chemical mechanisms. For this reason, we have implemented a new version of STOPS (v1.5) into CMAQ v5.0.2, which can be utilized with recent emissions inventories, improved chemical mechanisms and useful analyzing tools for the better simulation of Asian dust events.

The primary purpose of this study is to characterize underestimated PM concentrations simulated by standard CMAQ and determine the primary reason why CMAQ does not accurately capture PM peaks, particularly during the Asian dust events. We will introduce a new modeling framework using STOPS v1.5 (STOPS, hereafter) as an alternative to full CMAQ modeling and show that it enhances the performance of PM prediction to capture the severe dust storms over the Korean Peninsula. We will attempt to utilize STOPS for PM modeling with real-time input data (e.g., initial and boundary conditions and emission estimates) that allow STOPS to take into account the mostly updated input data inside of the modeling domain. We will run several STOPS simulations using timely reported PM concentrations based on real-time satellite observations, use remote-sensing data from the Geostationary Ocean Color Imager (GOCI) sensor to constrain PM concentrations (by injecting extra PM emissions) for STOPS, and investigate whether the constrained PM concentration produce more accurate PM simulations in STOPS. Then we will compare the results to corresponding surface observations and ultimately conclude by proposing the STOPS forecasting/modeling system as an effective tool for capturing PM forecasting/modeling over the East Asia, particularly in Korea.

## 2 Methodology

## 2.1 STOPS

STOPS is a hybrid Eulerian-Lagrangian-based modeling tool derived from the CMAQ model. A STOPS domain is a small sub-domain of a full CMAQ domain that moves along with the mean wind in its domain. Since STOPS inherits meteorological fields and initial and boundary conditions from a "host" CMAQ simulation, the movement of a STOPS domain is limited to the domain of the host CMAQ simulation. STOPS has the same vertical structure and physical and chemical processes as in CMAQ model, but it does not fully calculate advection term unlike full CMAQ model (Czader et al., 2015). The movement of the STOPS domain is determined by averaging the u and v wind components in the center column from the bottom layer up to the planetary boundary layer (PBL) height, weighted by the layer thickness. The averages of the u and v components are calculated by the following equations (Eq. (1)-(2)):

$$\bar{u} = \frac{1}{\sum_{L=1}^{PBLH} \Delta\sigma_F(L)} \sum_{L=1}^{PBLH} u_L \cdot \Delta\sigma_F(L) \tag{1}$$

$$\bar{v} = \frac{1}{\sum_{L=1}^{PBLH} \Delta\sigma_F(L)} \sum_{L=1}^{PBL\,\square} v_L \cdot \Delta\sigma_F(L) \tag{2}$$

where $\sigma_F = 1 - \sigma$ and $\sigma$ is scaled air pressure in a sigma coordinate system (dimensionless) defined as follows (Eq. (3)):

$$\sigma = \frac{(p - p_t)}{(p_s - p_t)} \tag{3}$$

where $p$, $p_t$, and $p_s$ denote air pressure at the current level and the top and surface levels of the model, respectively. Czader et al. (2015) presents more details on the basics of STOPS and the results of the application. The first version of STOPS (v1.0) was based on CMAQ v4.4 (Czader et al., 2015), but for this study, STOPS has been updated to v1.5, which is based on CMAQ v5.0.2.

## 2.2. Modeling system and experimental design

In this study, we used the CMAQ (v5.0.2) model (Byun and Schere, 2006). We configured the model so that it consisted of a single domain with a grid resolution of 27 km (174 × 128) covering the northeastern part of Asia (Fig. 1) with 27 vertical layers extending from the surface to 100 hPa. This CMAQ domain, which was set slightly larger than standard domain for East Asia study suggested by the

Geoscientific Model Development Discussions Open Access EGU

Clean Air Policy Modeling System (CAPMOS) (http://capmos.nier.go.kr/index.jsp) of the National Institute of Environment Research (NIER) in Korea, covers more areas of Gobi Desert, a major source of Asian dust. We used the CB05 and AERO6 for gas-phase and aerosol chemical mechanisms and obtained initial and boundary conditions from the standard CMAQ profile.

Anthropogenic emissions for the CMAQ domain were obtained from the MIX emissions inventory in 2010 (Li et al., 2015). This inventory contains gridded (0.25° × 0.25°) emissions information for black carbon (BC), carbon monoxide (CO), carbon dioxide ($CO_2$), nitrogen oxides ($NO_x$), ammonia ($NH_3$), organic carbon (OC), fine and coarse particulate matter ($PM_{2.5}$ and $PM_{10}$), sulfur dioxide ($SO_2$) and non-methane volatile organic compounds (NMVOC). To acquire high-resolution (1km × 1km) anthropogenic emissions in Korea, this study also refer to the Clean Air Policy Support System (CAPSS) emissions inventory in 2011 of the NIER (Lee et al., 2011a). The CAPSS inventory contains area, line, and point sources of CO, $NH_3$, $NO_x$, sulfur oxides ($SO_x$), total suspended particles (TSP), $PM_{10}$, and VOC. The emissions for the CMAQ simulations were prepared by the Sparse Matrix Operator Kernel Emissions (SMOKE) (Houyoux et al., 2000) system.

We simulated meteorological fields using the Weather Research and Forecast (WRF, v3.7) model (Skamarock et al., 2008) and used the 1°× 1° Final Operational Global Analysis (FNL) data of the National Centers for Environmental Prediction (NCEP) to determine the initial and boundary conditions for the simulation. To enhance the performance of WRF modeling, we applied an efficient data assimilation method (i.e., grid analysis nudging) to the WRF simulation. Several studies have reported on the benefit of grid analysis nudging to air quality modeling (Liu et al., 2012; Otte, 2008a; Otte, 2008b). To improve the accuracy of meteorological fields, we adopted the optimized grid analysis nudging options suggested by Jeon et al. (2015) for the East Asia simulations.

The time period for the WRF-CMAQ simulations was February 2015, when three days of Asian dust events, listed in Table 1, occurred in Korea. The model simulations lasted 38 days (January 21 to February 28, 2015), including the first ten days for spin-up.

### 2.3 In-situ and satellite measurements

This study referred to surface observational data from the air quality monitoring station (AQMS) network operated by NIER. The network measures real-time air pollutant concentrations and provides hourly concentrations for CO, $NO_2$, $O_3$, $PM_{2.5}$, $PM_{10}$, and $SO_2$. We gathered the measured $PM_{2.5}$ and $PM_{10}$ data in 2015 from the AQMS network to evaluate the modeled results. We also employed the aerosol optical depth (AOD), measured by a GOCI sensor from the geostationary orbit onboard the Communication Ocean and Meteorological Satellite (COMS). The GOCI level 1B (L1B) data provide hourly daylight spectral images (09:30-16:30 LST, 8 times a day) for East Asia. The spatial coverage extends to 2500 km × 2500 km centered at 36° N, 130° E with a 500 m resolution (Lee et al., 2010;

Choi et al., 2016). The 550 nm AOD data with a 6 km resolution were obtained from GOCI L1B data, which were based on a retrieval algorithm introduced by Choi et al. (2016). The GOCI-derived AOD data were used for constraining of PM concentration and the model evaluation.

# 3 $PM_{10}$ simulation results from standard CMAQ

## 3.1 Comparison with surface measurement

We simulated $PM_{10}$ concentrations by standard CMAQ and compared them with surface observational data obtained from the AQMS network of NIER in Korea. For this comparison, we selected 20 AQMS sites, evenly distributed in Korea (Fig. 1), and recorded mean $PM_{10}$ concentrations at all of the sites. We do not present the results for $PM_{2.5}$ because the simulated $PM_{2.5}$, similar to $PM_{10}$, exhibited almost same temporal variation and lower concentrations. In addition, the coarse particles comprise a major portion of the total PM during the Asian dust period, as described by Chun et al. (2001). From the comparison, shown in Fig. 2, the concentration of CMAQ-simulated $PM_{10}$ was slightly underestimated, but its temporal variation showed reasonably close agreement with observation except for the Asian dust episode (22-24 February). The CMAQ failed to capture the high peaks of $PM_{10}$ in the episode caused by the transport of massive dust from the Gobi Desert and Mongolia region. Table 2 shows statistical parameters for the simulated $PM_{10}$ concentrations. The performance of the simulated concentrations for the entire simulation period (February 2015) was poor. For example, the high and low values of RMSE (78.03 $\mu g/m_3$) and IOA (0.36) and the negative value of MBE (-39.94 $\mu g/m_3$) indicated that the CMAQ underestimated $PM_{10}$, and its temporal variation did not agree well with observation.

The calculated statistics for the period excluding the Asian dust episodes was much better than those for the entire period (Table 2). The large differences in these findings clearly reveal that the performance of CMAQ is relatively accurate for the regular simulation period, but it is not for the Asian dust period. As shown in Fig. 3, meteorological fields such as temperature and wind speed showed close agreement with observations, even during the Asian dust period. It suggests that the underestimated $PM_{10}$ concentration was not caused by the uncertainty of the simulated meteorology, but resulted from faulty estimation of dust emissions for the CMAQ simulation.

To enhance the performance of CMAQ for $PM_{10}$ simulations during the Asian dust period, we employed the in-line windblown dust module in the CMAQ v5.0.2. The employment of the in-line windblown dust module in CMAQ simulations did not provide discernible enhancement in $PM_{10}$ concentrations (Table 2) because of lower friction velocity than the threshold in the module during the simulation period (February 2015) (Table S1 in the supplementary document). This research also implies that more studies that enhance the capability of dust modules during the winter period should be performed.

### 3.2. Comparison with satellite-based observation

To evaluate the horizontal features of CMAQ simulated $PM_{10}$, we used GOCI-derived AOD. We converted the concentration unit in CMAQ to AOD for a fair comparison of the results with GOCI. The aerosol properties from the CMAQ simulation (CMAQ-derived AOD) were obtained by the following equations (Eq. (4)-(6)), introduced by Roy et al. (2007):

$$AOD_{CMAQ} = \sum_{i=1}^{N} (\sigma_{sp} + \sigma_{ap})_i \Delta Z_i \tag{4}$$

$$\sigma_{sp} = (0.003) f_t(RH)[NH_4{}^+ + SO_4{}^- + NO_3{}^-] + (0.004)[OM] + (0.001)[FS] + (0.0006)[CM] \tag{5}$$

$$\sigma_{ap} = (0.01)[LAC] \tag{6}$$

where $i$ is the vertical layer number, $\Delta Z$ is the layer thickness, and the brackets indicate mass concentrations in mg/m$^3$ units. The OM, FS, CM, and LAC denote mass concentrations of organic species, fine soil, coarse particles, and light-absorbing carbon, respectively. The specific scattering coefficients in the equations (i.e., 0.003, 0.004, 0.001, 0.0006, 0.001) are represented in units of m$^2$/mg. The $f_t(RH)$, calculated by the method described by Song et al. (2008), denotes relative humidity based on the aerosol growth factor.

Figure 4 represents a comparison of time-averaged AOD derived from GOCI and CMAQ. For fair comparison of their AOD, we removed grid cells from GOCI data consisting of fewer than 15 pixels (i.e., bad pixels) because of cloud contamination and corresponding grid cells in CMAQ. In GOCI-derived AOD, several blank areas appeared in the northern part of the Korean Peninsula near the northeastern region of China and in most regions of Japan because of the significantly high fraction of clouds over these areas. The horizontal features of the CMAQ-derived AOD were similar to those of the GOCI-derived AOD, but CMAQ overestimated the AOD near the southeastern part of China. On the other hand, compared to the GOCI-derived AOD, the CMAQ underestimated the AOD over the Yellow Sea and Korea. As mentioned in Sect. 3.1, CMAQ underestimated $PM_{10}$ concentrations in Korea. The CMAQ-derived AOD in Korea, compared to GOCI-derived AOD, was also underestimated, consistent with the surface measurements. Two comparisons using the satellite and surface measurements indicated the same results that the CMAQ barely captured the high levels of PM in Korea during the simulation period in this study (February 2015). The discrepancy between CMAQ- and GOCI-derived AOD is primarily due to uncertainty present in PM precursor emissions (Jeon et al., 2015) because the meteorology used for the CMAQ simulation exhibited high accuracy (Fig. 3).

Compared to the GOCI-derived AOD, the CMAQ-derived AOD near the northern regions of

the Korean Peninsula was underestimated. This underestimation may have resulted from the failure of CMAQ to simulate the breakout and loading of Asian dust and its transport to the Korean Peninsula on 22-24 February. The CMAQ-derived AOD was underestimated primarily in the moving pathway of the Asian dust (i.e., between the Gobi Desert (source area) and Korean Peninsula (receptor area)). As addressed in Sect. 3.1, the in-line windblown dust module in CMAQ failed to accurately estimate the dust emissions during the Asian dust period and it caused the underestimated AOD near the northern regions of the Korean Peninsula.

To further investigate the issue of underestimation of CMAQ during the period of Asian dust (Table 1), we compared the GOCI- and CMAQ-derived AODs on each event day. Unfortunately, the comparison was available only on 22 February since the GOCI-derived AOD included a significantly high number of blank pixels on the other event days because of the high fraction of clouds cover. Figure 5 shows GOCI- and CMAQ-derived daily mean (09:30-16:30 LST) AODs on 22 February. The GOCI-derived AOD clearly showed massive dust near the northwestern regions of the Korean Peninsula and the eastern part of China and densely distributed dust particles over the Yellow Sea that were transported from the Gobi Desert. In contrast, CMAQ did not reproduce the high amounts of dust particles near the Korean Peninsula primarily because of the failure of the in-line windblown dust module (see the details, Table S1 in the supplementary document).

We conclude that CMAQ clearly underestimated $PM_{10}$ concentrations during the simulation period and failed to capture peaks during the Asian dust period starting on 22 February. Thus, to enhance the performance of standard CMAQ, we attempted to utilize STOPS for the $PM_{10}$ simulation. To capture the dust enhanced $PM_{10}$ in Korea (receptor region), we can use the dust storm data temporarily detected by satellite measurements between the source and receptor regions as an input for the STOPS modeling. The following sections will describe, in detail, the STOPS modeling system and its application results.

# 4 Application of STOPS for $PM_{10}$ prediction

## 4.1 Configuration of STOPS

The configuration of the CMAQ sub-domain for the base STOPS simulation consists of 61 × 61 horizontal grid cells that covers a portion of the Korean Peninsula and the Yellow Sea, and its initial position was near the northern part of the Yellow Sea (40° N, 119° E) (Fig. 1), the transporting pathway of Asian dust. The simulated $PM_{10}$ concentrations of base STOPS (without constrained $PM_{10}$ concentrations) during Asian dust events (22-24 February) closely agreed with those of CMAQ (Fig. S1 in the supplementary document). The correlation coefficients (R) for each day were 0.94, 0.96, and 0.97,

Geoscientific Model Development Discussions Open Access EGU

indicating that the results from STOPS and CMAQ are significantly correlated. This reasonable consistency of STOPS and CMAQ results justifies the use of STOPS instead of CMAQ, in this study.

### 4.2 $PM_{10}$ forecasting using STOPS

Assuming that the CMAQ $PM_{10}$ simulation results in this study were used for forecasting purposes, the severe dust events starting on 22 February could not be predicted; that is, the forecasting for the Asian dust events would have failed. Thus, to accurately forecast the transport of massive dust storm, we must take into account the most recent and accurate input data. Figure 6 shows the GOCI-derived AOD on 21-22 February, when a dust storm was approaching Korea (receptor region) according to the GOCI measurements. The massive dust storm was not evident from the GOCI-derived AOD on 21 February, but a clear core of the dust storm in the northwestern region of the Korean Peninsula was first seen at 10:30 LST on 22 February. For an accurate $PM_{10}$ prediction, we should conduct a new CMAQ forecasting run with updated data from the GOCI-derived AOD in near real-time to update the current forecasting results. However, the new forecasting using the CMAQ with updated input cannot be provided within a short time because of its long simulation time (i.e., 5-6 hours for a two-day forecasting run). STOPS, however, can be used in this situation because of its very short simulation time and its similarity to CMAQ in performance, shown in Sect. 4.1. Upon observation of the dust core from the GOCI-derived AOD at 10:30 LST on 22 February, an updated $PM_{10}$ forecasting using STOPS with real-time AOD data can be performed in a short time (i.e. a few minutes) and the current forecasting results can be replaced by the results from the updated STOPS. For the updated $PM_{10}$ forecasting using STOPS, we will use the GOCI-derived AOD as new initial and boundary chemical conditions for $PM_{10}$ species in the same simulation time (10:30 LST, February 22). However, the approach does not fully consider all transport of dust from a source region (note: dust storms are usually discovered between the source and receptor regions from remote sensing or in-situ surface measurements). The impact of the updated initial and boundary chemical conditions on the STOPS domain would be mitigated within a few hours. Thus, to make the best use of the AOD data, we attempted to utilize the GOCI-derived AOD data to constrain PM concentrations for the updated STOPS run.

#### 4.2.1 Satellite-adjusted PM concentrations

To provide the updated PM concentrations that take the real-time AOD into account, we constrained the standard PM concentration using the GOCI-derived AOD data at the beginning of the updated forecast. For the constraint, we first attempted to directly add the extra amount of PM, which was estimated from the GOCI-derived AOD, to the current PM concentrations simulated by standard CMAQ. However, the sudden and rapid changes in PM concentration made the CMAQ simulation unstable and they sometimes caused unexpected termination of CMAQ runs due to overflow error. To resolve this

problem, we regarded the extra amount of PM estimated from the GOCI-derived AOD as alternative emissions and indirectly constrained the original PM concentrations by using alternative emissions into standard emission. The GOCI-derived AOD was converted to emission unit and the converted emission values were used. We should note that the alternative emissions are not real, but the enhanced amount of dust particles which are taking the form of emission. We concluded this methodology could be an effective way to reflect the satellite measured AOD to CMAQ simulation without possible computational error.

As indicated in Fig. 6, the massive dust storm was first captured by the GOCI-derived AOD at 10:30 LST on 22 February, so we adjusted the standard emissions at a corresponding time based on the GOCI-derived AOD and used them for the updated forecast using STOPS. We should note that the AOD and the emissions rate are expressed in different units; the AOD is a unitless value, while the emissions rate is expressed in units of grams per second (particles) or moles per second (gas-phase species); therefore, we employ a scaling factor to convert the AOD to the emissions rate. To find a reasonable scaling factor, we re-gridded the domain of the high GOCI-derived AOD data so that it corresponded to the CMAQ domain and compared the AOD in each grid cell with corresponding emission rates of total PM in the MIX inventory (e.g., $PM_{10}$). We used only the grid cells with valid AODs (no missing values) and emission rates (> 0) for the comparison and then calculated the average ratio of the AOD to emissions rates. The calculated ratio was 1,884.49 g $s^{-1}$ for this case, indicating that AOD inside the modeling domain was 1,884.49 times as large as the emissions rate of total PM. It should be noted that the ratio cannot generally explain the relationship between AOD and emissions. Because the relationship is valid for only a particular domain (Fig. 1) and time (10:30 LST on 22 February, 2015), the ratio for each case should be recalculated.

For the unit conversion from the AOD to the emissions rate of total PM, we used the estimated ratios as scaling factors and obtained the alternative emissions for the updated STOPS forecasting by the following equation (Eq. (7)):

$$PMT_{i,j} = AOD_{i,j} \times \mathrm{SF} \quad (7)$$

where $PMT_{i,j}$ and $AOD_{i,j}$ represent the emission rate of total PM and GOCI-derived AOD in each grid cell, respectively. SF is the calculated scaling factor (1,884.49 g $s^{-1}$), which indicates the relationship between the AOD and the emissions rate.

For the CMAQ simulation, we split the calculated PMT into several specific species, including coarse and fine particles, used for the CB05-AERO6 chemical mechanism. For speciation, we investigated and determined the specific fractions of each PM species during the Asian dust events based on the findings in Kim et al. (2005) and Stone et al. (2011), which described the composition of measured PM during the Asian dust periods. After calculating and using the average fractions reported

in the two studies (Table 3) for the speciation of PMT, we split the adjusted PMT into specific PM species in CB05-AERO6 mechanism. More than half of the PMT was allocated to coarse particles (PMC) because they comprise a major percentage of Asian dust, as reported in several studies (Kim et al., 2003; Lee et al., 2004; Kim et al. 2005; Stone et al., 2011). Calculated alternative emissions were injected into standard PM emissions in each grid cell. Based on the findings by Kim et al. (2010), the amounts of the alternative emissions were assumed to be distributed below the altitude of 3 km (1 to 11 vertical layers).

Figure 7 presents comparisons between the standard and constrained $PM_{10}$ (by using alternative emissions) concentrations at the beginning time of the STOPS simulation. The $PM_{10}$ from standard CMAQ exhibited high concentration over the eastern part of China, central part of the Yellow Sea and northwestern part of the Korean Peninsula. By contrast, the constrained $PM_{10}$ by the alternative PM emissions (Fig. S2 in the supplementary document) exhibited significantly increased concentration, particularly in the northwestern part of the Korean Peninsula, southern part of the Yellow Sea and western part of the Korean Peninsula (Fig. 7). The constrained $PM_{10}$ concentration showed similar features as those of the GOCI-derived AOD, shown in Fig. 5-(a), implying that the dense dust attributed by Asian dust were accurately reflected in the STOPS simulation.

#### 4.2.2 Enhanced $PM_{10}$ forecasting using STOPS

We ran an updated $PM_{10}$ forecasting simulation using STOPS with the constrained PM concentration (by using alternative emissions) and examined the improvement in its accuracy over that of standard CMAQ. The STOPS simulations were assumed to cover one-day (24 hours) forecasting, which began at 11:00 LST on 22 February, immediately following the massive dust first observed in the GOCI-derived AOD between the source and receptor regions. We should note that the duration of the release of alternative emissions strongly affected the simulated $PM_{10}$. Hence, determining the duration of the release of the alternative emissions plays an important role in updating a forecast using STOPS, so we ran four sensitivity simulations with different release durations (3hr, 6hr, 12hr, and 24hr) using STOPS and compared all of the results with those of standard CMAQ and available $PM_{10}$ surface measurements. The comparison in Fig. 8 of the four simulations of observed and updated STOPS-simulated $PM_{10}$ concentrations exhibits clear differences in the temporal variation of $PM_{10}$ resulting from the impact of the durations. As addressed in Sect. 3.1, the standard CMAQ run failed to capture the drastic increase in $PM_{10}$ concentrations on 22 February because of the faulty estimation of transported Asian dust. The results of updated STOPS showed significant improvements over those of standard CMAQ. The results of the four updated STOPS simulations indicated higher $PM_{10}$ concentrations than those of CMAQ, and they were much closer to observations.

Interestingly, the four updated STOPS simulations exhibited noticeable differences of $PM_{10}$ time series according to variations in the duration of the release of the alternative emissions. Figure 8

shows that simulated $PM_{10}$ from STOPS with a duration of release of three hours (STOPS_E3) closely agreed with observations during the first three hours. However, the simulated $PM_{10}$ began to decrease immediately after the third hour, and the agreement with observations gradually worsened with time. The results of the STOPS simulations with different durations of release of 6, 12, and 24 hours (STOPS_E6, STOPS_E12 and STOPS_E24, respectively) were almost the same as those of STOPS_E3. In other words, the impact of the alternative emissions on the improved $PM_{10}$ simulation results, which was clear within their respective durations of release, was mitigated after the release ended. STOPS_E24 represented the closest agreement with observations, implying that STOPS_E24 produced the greatest improvement in one-day $PM_{10}$ forecasting because of continuously released updated emission during the entire forecasting time (24 hours).

Despite its positive performance in one-day $PM_{10}$ forecasting, STOPS_E24 did not perfectly capture the high $PM_{10}$ concentrations during the Asian dust event. In fact, it underestimated the peak of observed $PM_{10}$, which may have resulted from uncertainty inherent in the methodology using AOD estimation. Direct conversion from the AOD to the alternative emissions rate using a scaling factor is challenging because it has not yet proven reliable by existing studies. Hence, the uncertainty inherent in unit conversion might have contributed to the inaccuracy of the emissions rate. In addition, the GOCI-derived AOD data contained some blank cells resulting from the high fraction of clouds cover during the event on 22 February, and as a consequence, it did not accurately represent the distribution of transported Asian dust. The most probable reason for the underestimated $PM_{10}$ simulated by STOPS was that the alternative emissions during the first time step (11:00 LST on 22 February) were subsequently used for all of the time steps without accounting for spatiotemporal variations. Since the horizontal and vertical distributions of the Asian dust changed with time, the alternative emissions in the first time step did not accurately represent the varied dust distribution in the next time step. The uncertainty with regard to the alternative emissions definitely became larger as time passed. Indeed, $PM_{10}$ concentrations simulated by the updated STOPS showed close agreement with observations during the first six hours (Fig. 8), but agreement gradually widened with time. However, as updated data from observation in later hours cannot be reflected at the beginning of forecasting, such a problem is inevitable in a forecasting mode. Thus, repeated forecasting for short time periods (e.g., six hours) with the variable alternative emissions could possibly provide more accurate $PM_{10}$ results for the Asian dust events. STOPS would be very useful for repeated $PM_{10}$ forecasting simulations because of its remarkably short simulation time (a few minutes),

To verify the changed horizontal distribution of $PM_{10}$ resulting from the effect of constrained PM, we compared the simulated surface $PM_{10}$ concentrations from the updated STOPS to those from standard CMAQ. Figure 9 shows the horizontal distribution of surface $PM_{10}$ concentration inside of the STOPS domain simulated by standard CMAQ and STOPS_E24, which indicates the most accurate one-day forecasting results of all the STOPS simulations (from Fig. 8). The location of the STOPS domain moved slightly toward a southeasterly direction according to the changed mean wind in the domain. In

the first time step (0 hr, 11:00 LST, 22 February), STOPS_E24 showed the same $PM_{10}$ distribution as standard CMAQ because the initial condition for the STOPS simulation was provided by the standard CMAQ. After eight hours, the $PM_{10}$ concentration from STOPS_E24 differed from that of the standard CMAQ owing to the effect of the alternative emissions by the GOCI-derived AOD. After sixteen and twenty four hours, the difference became more pronounced. Results of standard CMAQ did not show a high level of $PM_{10}$, but those of STOPS_E24 showed a $PM_{10}$ concentration of at least 200 $\mu g\ m^{-3}$ near the Korean Peninsula. Specifically, they showed extremely high $PM_{10}$ concentrations of over 1,500 $\mu g\ m^{-3}$ in the northwestern part of the Korean Peninsula. Figure 6 (10:30 LST on 22) indicates massive dust over that area from the GOCI-derived AOD consistent with the enhanced $PM_{10}$ concentrations. The massive dust over the region were transported to Korea and led to significantly enhanced levels of $PM_{10}$. The horizontal distributions of $PM_{10}$ at higher vertical levels up to 3 km showed similar features at the surface layer because the alternative emissions were evenly distributed below that level.

Overall, even with the uncertainties addressed above, the massive dust storm near the Korean Peninsula on an Asian dust day was reasonably reproduced by the STOPS simulation with constrained PM by GOCI-derived AOD. These results indicate that the STOPS could possibly be used for new $PM_{10}$ forecasting with real-time constraint of PM concentration and this methodology should enhance the performance of $PM_{10}$ forecasting and modeling.

## 5 Summary

This study introduced a new modeling framework using a hybrid Eulerian-Lagrangian model (called STOPS) that showed almost the same performance as CMAQ, but with a shorter simulation time. STOPS v1.5 has been implemented into CMAQ v5.0.2 for $PM_{10}$ simulations over the East Asia during Asian dust events and we investigated possibility of using STOPS to enhance the forecasting performance of CMAQ. During the entire simulation period (February 2015), the standard CMAQ underestimated $PM_{10}$ concentrations compared to surface observations and it failed to capture the $PM_{10}$ peaks of Asian dust events (22-24 February). The accurately simulated meteorology implied that the significantly underestimated $PM_{10}$ concentration was not due to meteorology but instead to inaccurately estimated dust emissions for the CMAQ simulation. We also evaluated the horizontal feature of CMAQ simulated $PM_{10}$ using satellite-observed data (GOCI). The $PM_{10}$ results from the standard CMAQ run were compared to those of the GOCI-derived AOD and the results indicated that CMAQ barely captured the transported dust from the Gobi Desert to the Korean Peninsula during the Asian dust events.

To improve the underestimated $PM_{10}$ results from the CMAQ simulation, we used the STOPS model and ran several simulations using constrained PM concentrations (by using alternative emissions) based on the GOCI-derived AOD, which reflected the most recent initial and boundary conditions near

the Korean Peninsula. The STOPS simulations showed higher $PM_{10}$ concentrations than the standard CMAQ and indicated clear dependence on the duration of the alternative emission release. The STOPS simulations showed a $PM_{10}$ concentration very close to that of surface observational data, but they did not accurately reproduce the high $PM_{10}$ concentration during the Asian dust events primarily resulting from the inherent uncertainty of the methodology used for the constraining PM concentration. The direct conversion from the AOD to the alternative emission rate using a scaling factor was challenging because it has not yet proven reliable by existing studies. In addition, the GOCI-derived AOD data were missing many values because of the high fraction of clouds cover during the event and consequently, it did not accurately reflect the massive dust storm on the Asian dust day, which contributed to the underestimated $PM_{10}$.

Overall, STOPS successfully reproduced the high level of $PM_{10}$ over the Korean Peninsula during the Asian dust event with constrained PM concentration using satellite measurements. Although STOPS indicated significantly high $PM_{10}$ enhancement for the episode, it still requires improvement before its results can be generalized. Thus, we should direct our study toward additional verification of the methodology regarding on unit conversion and numerous sensitivity simulations for different cases to determine the optimal duration of the release of the alternative emissions. The results of this study are an ideal starting point for such studies.

The ultimate goal of this study was to suggest an effective tool for successive $PM_{10}$ forecasting and modeling over the East Asia, and the results clearly showed the reliability and various advantages of STOPS modeling. Therefore, because of its reliable performance with remarkably high computation efficiency, the STOPS model could prove to be a highly useful tool for enhancing $PM_{10}$ forecasting/modeling performance over the East Asia. Further, the benefit of STOPS modeling could be generalized to the forecasting and modeling of unexpected events such as wildfires and upset emissions event in industrial regions.

## Code availability

The STOPS v1.5 source code can be obtained by contacting the corresponding author at ychoi6@uh.edu.

## Acknowledgments

This study was partially supported by funding from the Ministry of Environment of Korea (NIER-1900-1946-302-210). We thank the team members of GOCI and AQMS for the preparation of the remote sensing data and PM measurement data, respectively.

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

Table 1. Observed $PM_{10}$ and $PM_{2.5}$ concentrations ($\mu g\ m^{-3}$) recorded on each days of an Asian dust event in February 2015. The values are averaged of the 20 AQMS sites shown in Fig. 1. D_Max denotes daily maximum concentrations and D_Mean daily mean concentrations.

| | $PM_{10}$ | | $PM_{2.5}$ | |
|---|---|---|---|---|
| | D_Max | D_Mean | D_Max | D_Mean |
| Feb 22 | 345.47 | 111.52 | 28.75 | 18.85 |
| Feb 23 | 472.47 | 341.63 | 72.67 | 43.61 |
| Feb 24 | 175.88 | 111.86 | 37.78 | 23.46 |

Table 2. Statistical parameters of $PM_{10}$ concentrations at 20 AQMS sites in Korea for the simulations without the dust module (CMAQ), with the in-line dust module (CMAQ_Dust).

| | Entire period | | | Without Dust Events | | |
|---|---|---|---|---|---|---|
| | RMSE | IOA | MBE | RMSE | IOA | MBE |
| CMAQ | 78.03 | 0.36 | -39.94 | 28.56 | 0.81 | -22.83 |
| CMAQ_Dust | 78.03 | 0.36 | -39.94 | 28.56 | 0.81 | -22.83 |

Table 3. Specific fractions (%) for the splitting of total PM emission into specific PM species in the CB05-AERO6 chemical mechanism used in this study.

| PM Emission Species | Fraction | PM Emission Species | Fraction |
|---|---|---|---|
| PMC (Coarse Particle) | 55% | PCA (Calcium) | 2% |
| PMOTHR (Unspeciated $PM_{2.5}$) | 25% | PEC (Elemental Carbon) | 1% |
| $PSO_4$ (Sulfate) | 8% | PNA (Sodium) | 1% |
| $PNO_3$ (Nitrate) | 3% | PCL (Chloride) | 1% |
| POC (Organic Carbon) | 3% | PK (Potassium) | 1% |
| PNH4 (Ammonium) | 2% | | |

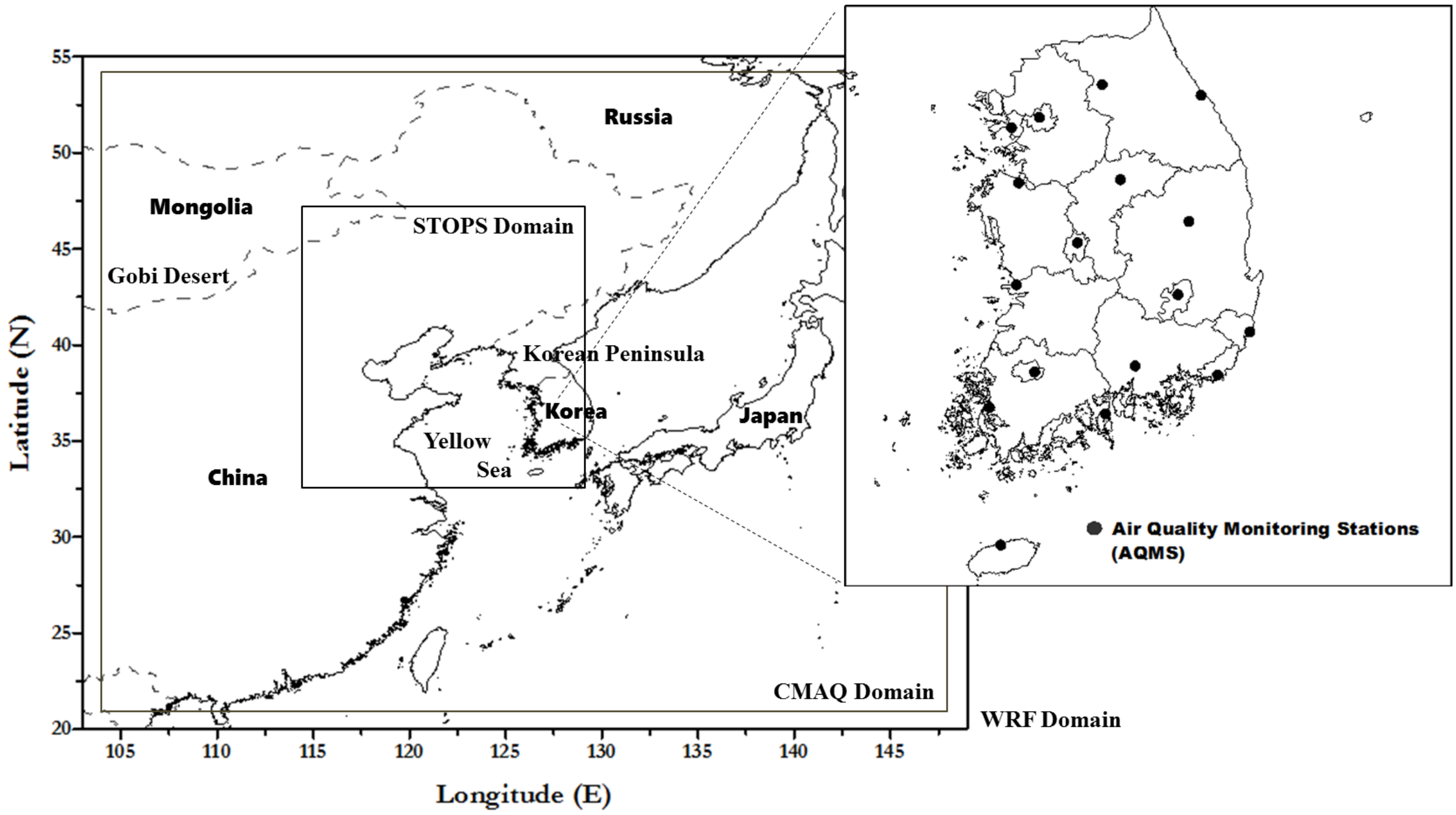

Figure 1. Domains for the WRF and CMAQ modeling. The right panel shows the location of the air quality monitoring stations (AQMS) used in this study.

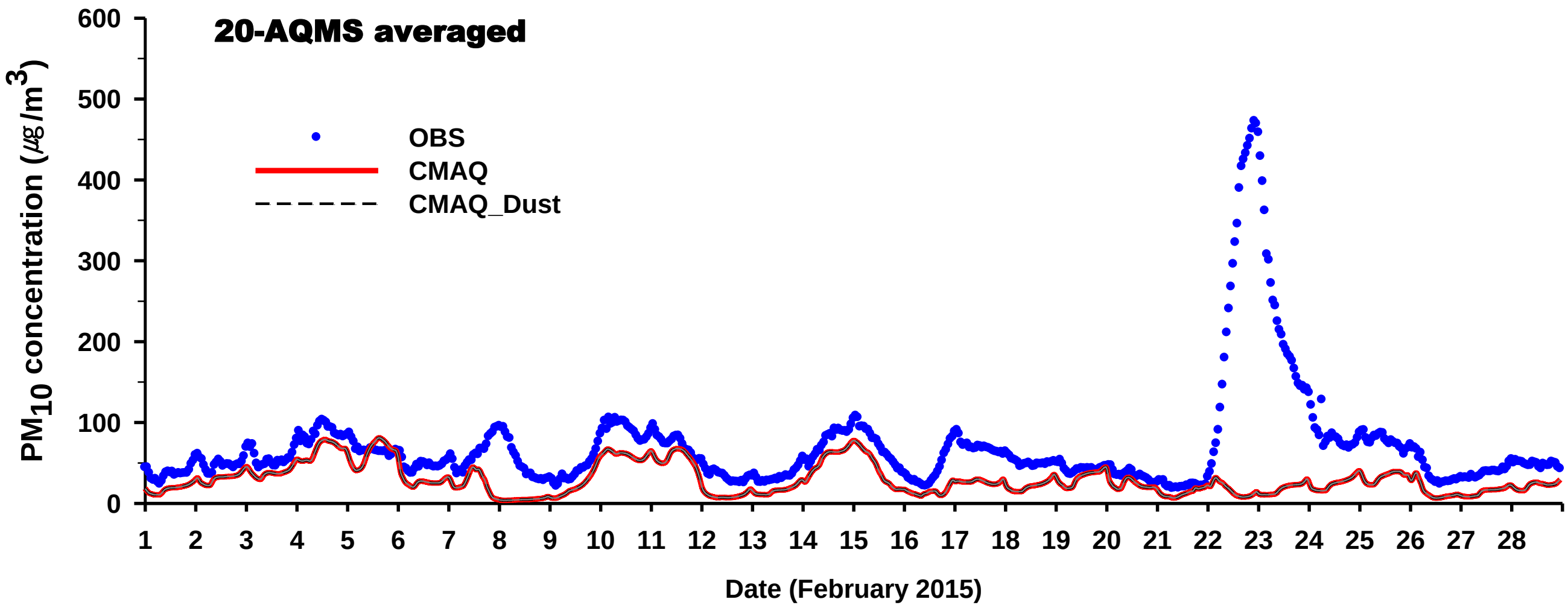

Figure 2. Time series of observed (OBS, blue dots) and simulated (CMAQ: red line, CMAQ_Dust: black dashed line) $PM_{10}$ concentrations in February 2015. The values are averaged values for 20 AQMS sites: CMAQ_Dust is closely coupled with the standard CMAQ modeling results (red line).

(a)

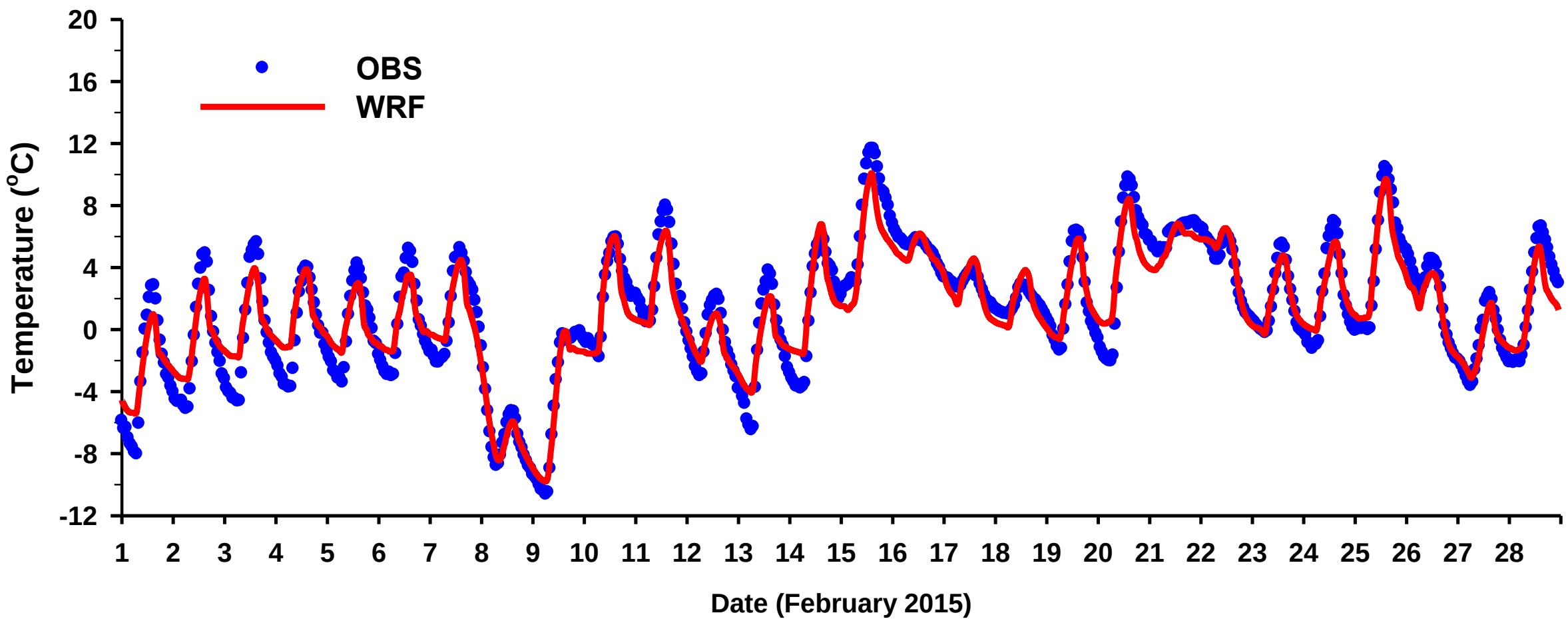

(b)

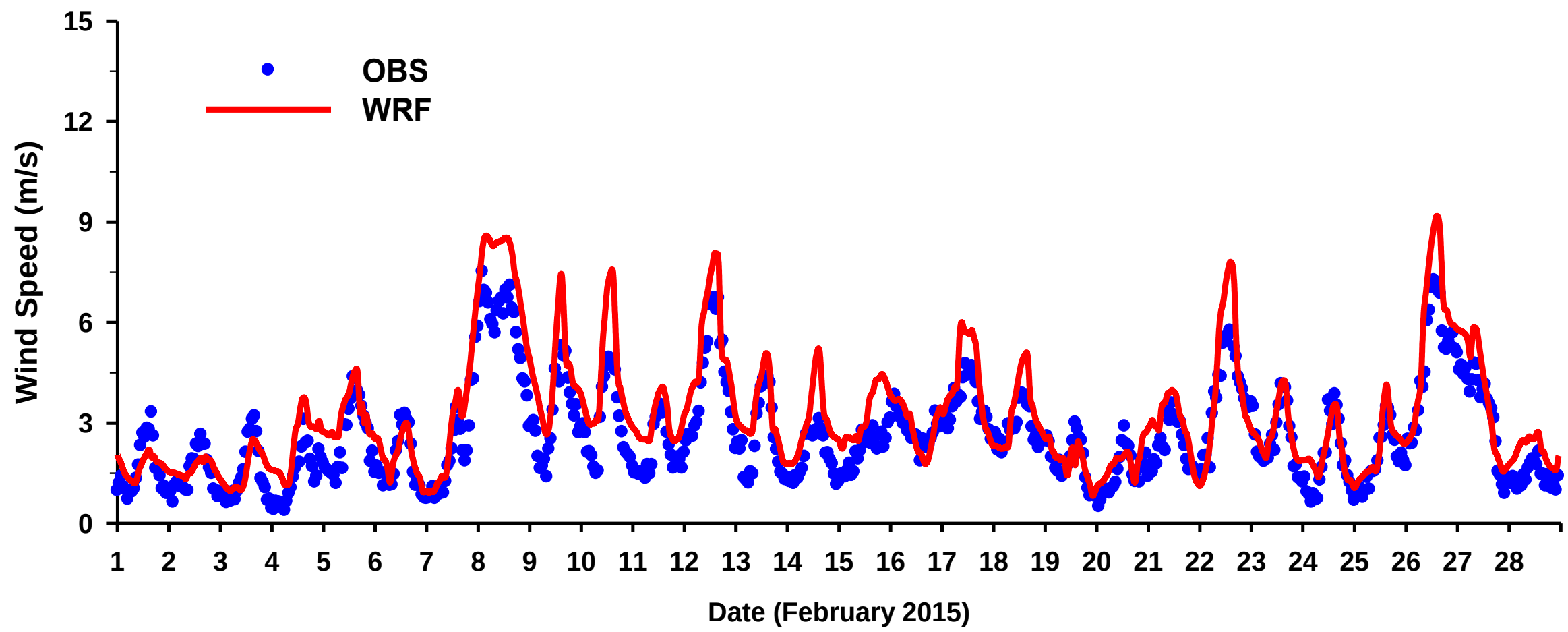

Figure 3. Time series of observed (OBS, blue dots) and WRF simulated (WRF, red line) (a) temperature and (b) wind speed in February 2015. The values are averaged values for 20 AQMS sites.

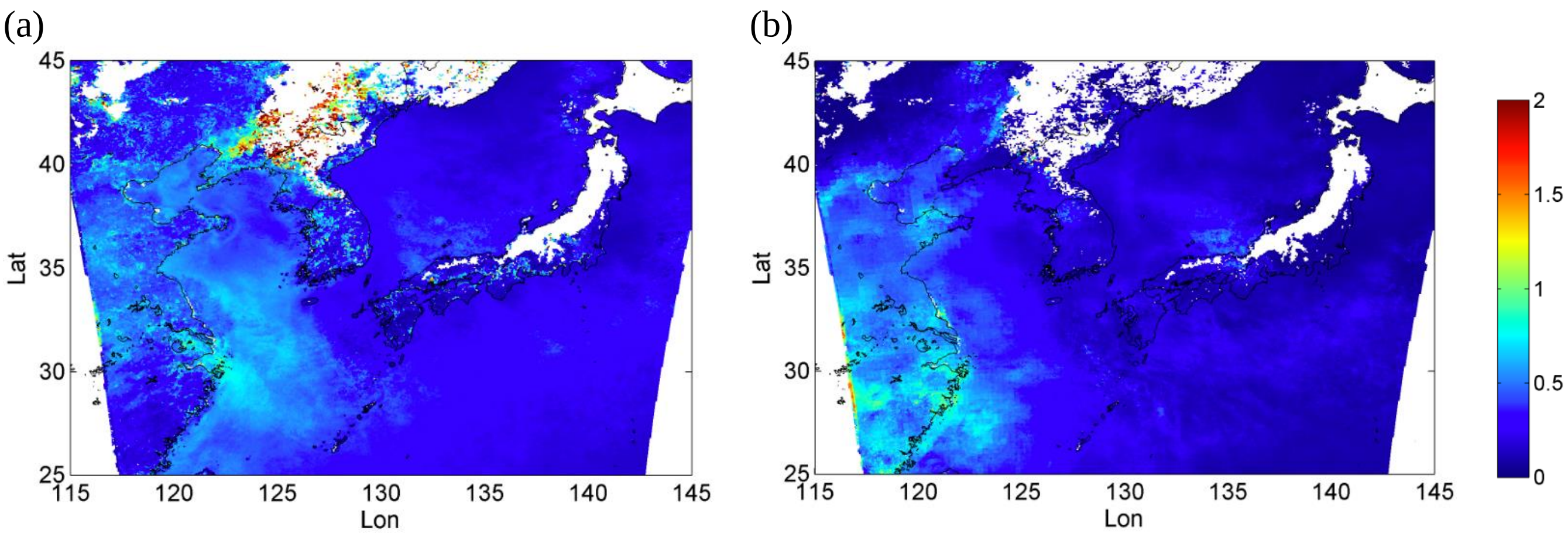

Figure 4. The (a) GOCI- and (b) CMAQ-derived AOD (550 nm) during the entire time period of simulations. The values are averaged for February 2015.

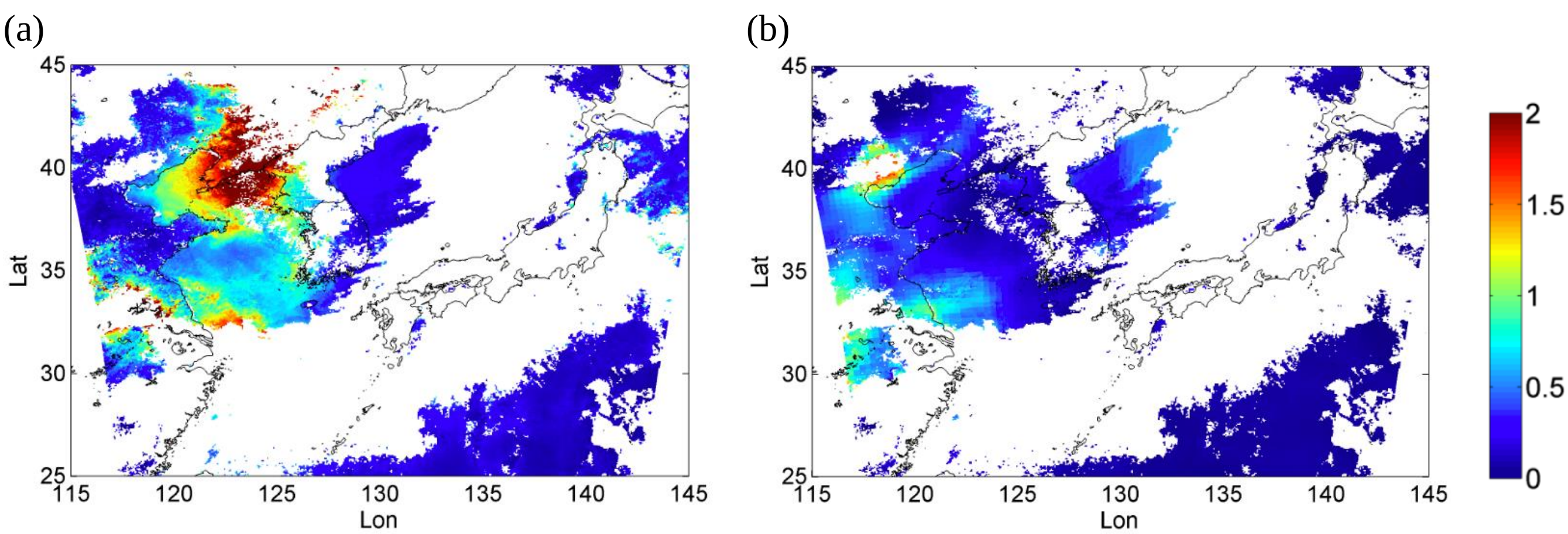

Figure 5. The (a) GOCI- and (b) CMAQ-derived AODs (550 nm) on 22 February. The values are averaged from 09:30 to 16:30 LST.

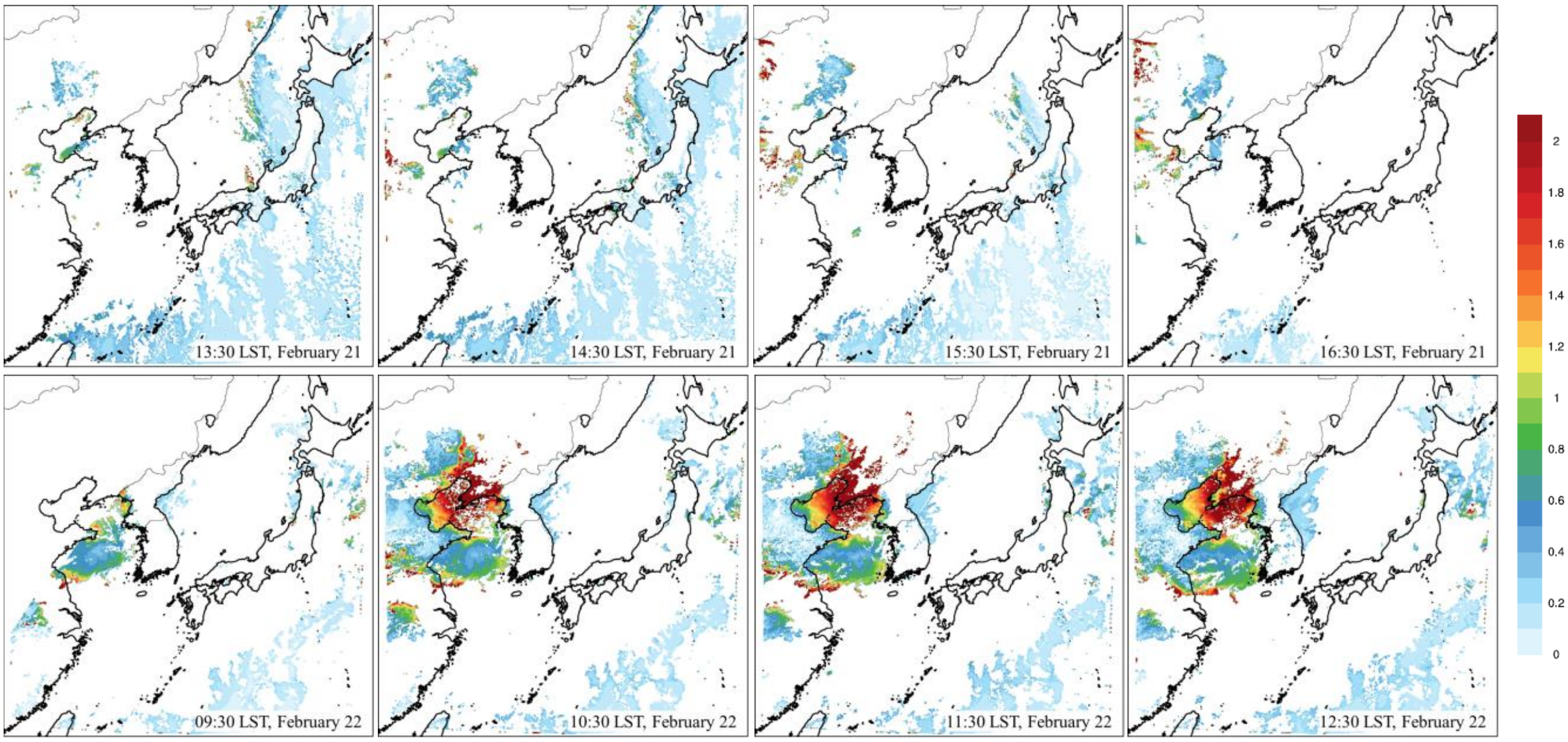

Figure 6. The GOCI-derived AOD (550 nm) from 13:30 LST on 21 February to 12:30 LST on 22 February, 2015.

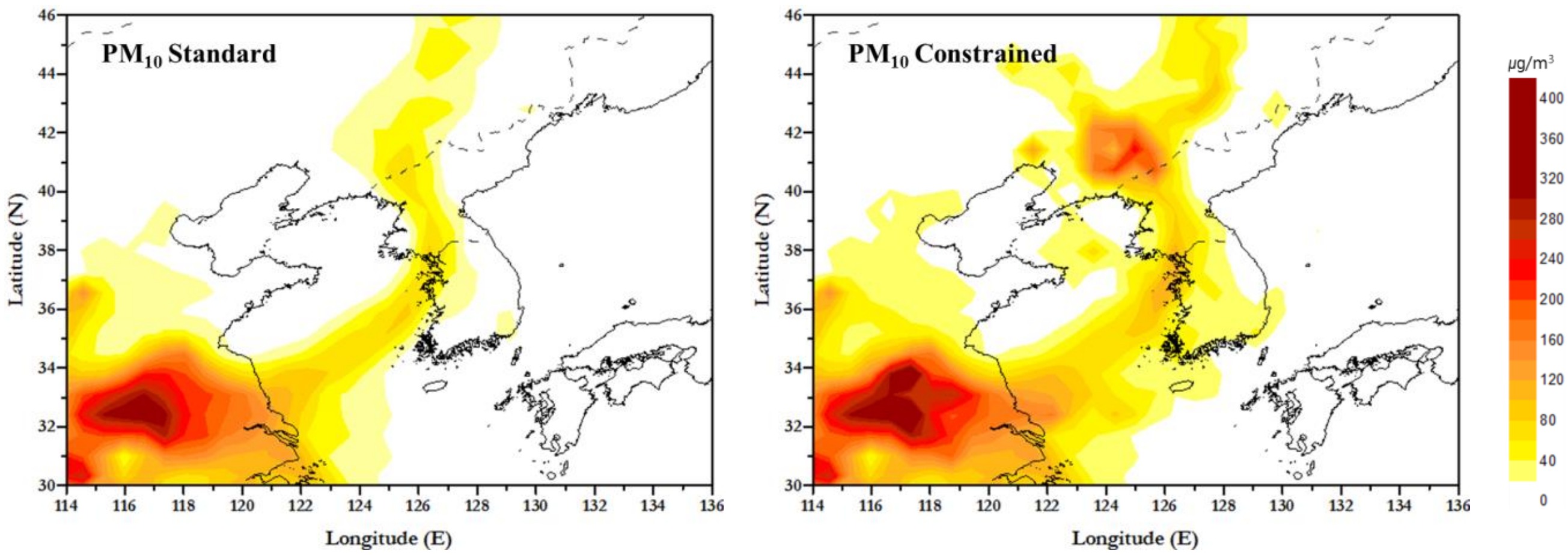

Figure 7. Difference between the $PM_{10}$ concentrations ($\mu g\ m^{-3}$) of standard and constrained CMAQ runs at 12:00 LST on 22 February. The constrained CMAQ run denotes the CMAQ simulation with alternative emissions for representing the estimation of the GOCI-derived AOD.

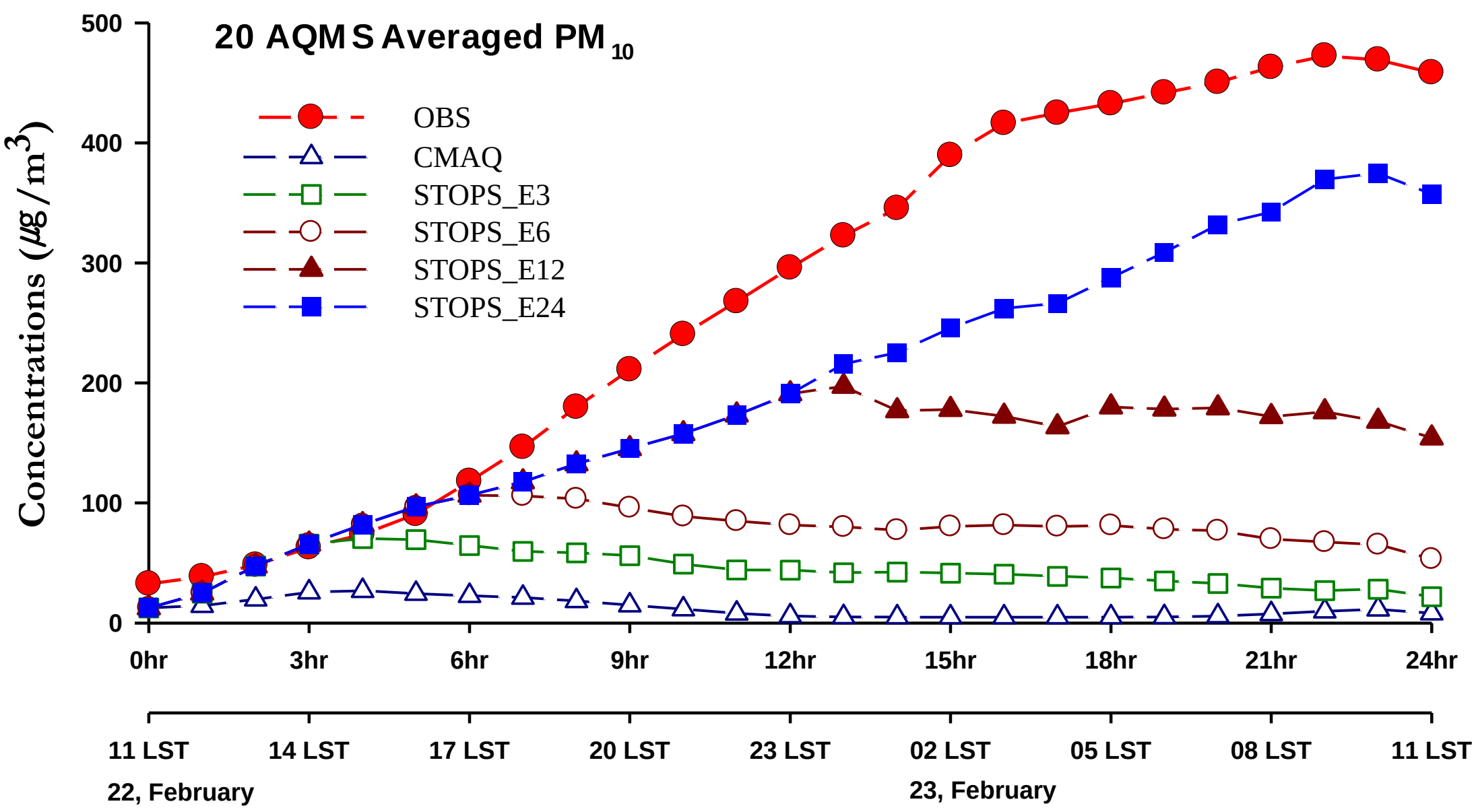

Figure 8. Comparison of observed, CMAQ-simulated and STOPS-simulated $PM_{10}$ concentrations during the 24 hours from 10:00 LST on 22 February, 2015.

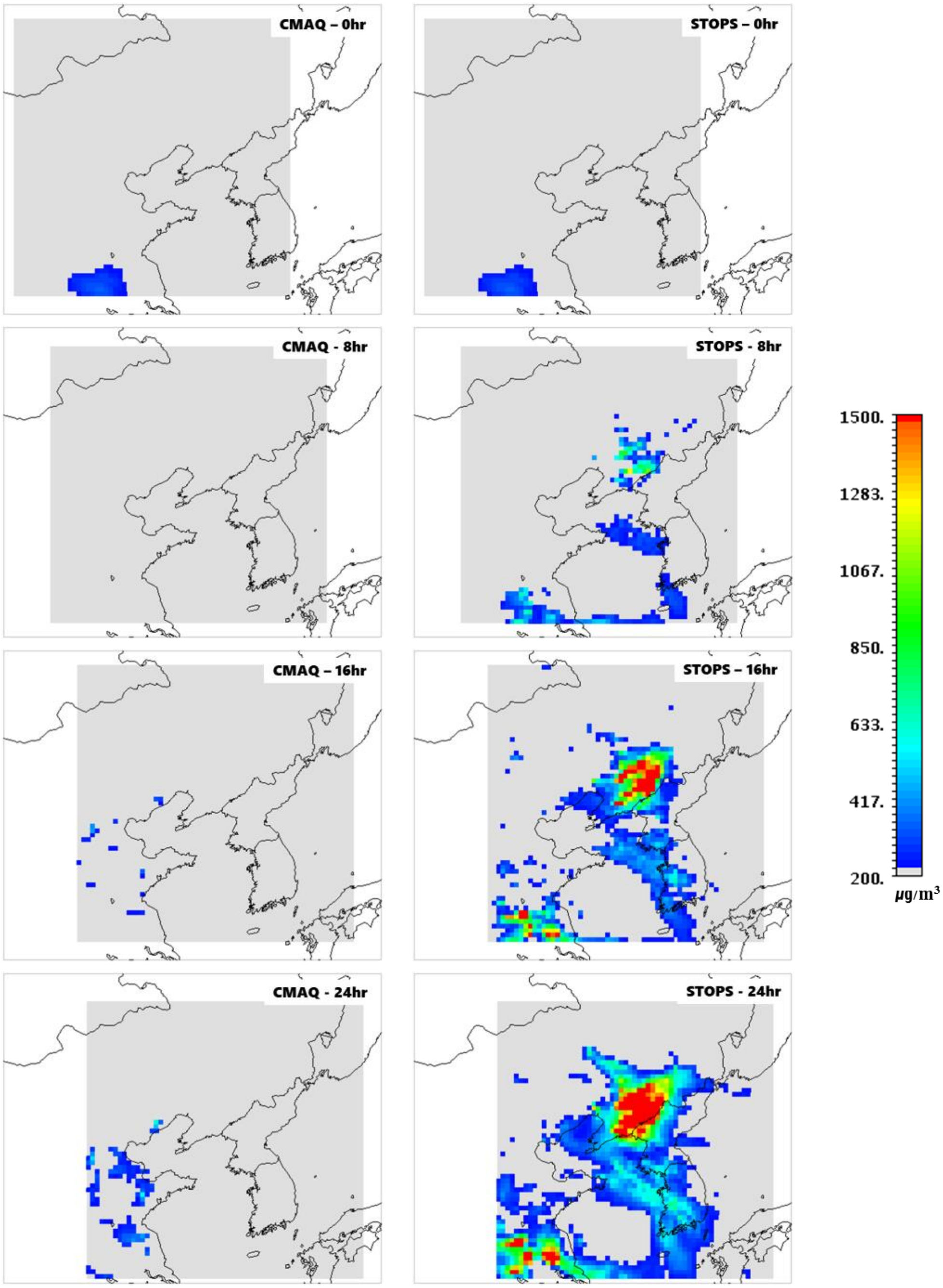

Figure 9. Horizontal distributions of standard CMAQ- and STOPS_E24-simulated surface $PM_{10}$ concentrations inside the STOPS domain. The concentrations were recorded at eight-hour intervals after the beginning of the simulation (11:00 LST on 22 February).