# Peer review of "Computationally efficient air quality forecasting tool: implementation of STOPS model into CMAQ v5.0.2 for a prediction of Asian dust"

_Geoscientific Model Development, 2016_

## Referee Comment (RC1) · Anonymous Referee #1 · 19 Aug 2016

***Please see the supplementary PDF for a better version ****

Reviewing "The new implementation of a computationally efficient modeling tool into CMAQ and its application for a more accurate prediction of Asian Dust" by Jeon et al.

This study by Jeon et al. implemented a new hybrid Lagrangian-Eulerian model, STOPS, into CMAQ, to improve the air quality forecasting. Jeon et al. use the STOPS modeling framework with constrained PM from geostationary satellite AOD to improve the Asian dust event that occurred in South Korea on Feb 22-24, 2015. It demonstrates well how STOPS could be useful in air quality forecast, particularly for the unusual air

quality events such as Asian dust transport. The merit of using STOPS is on low computational burden compared to CMAQ, which can be critical for emergency forecasting.

The manuscript is well within the scope of GMD. However, the manuscript requires some revisions. Please see my comments below. In addition to those comment, I believe science writing in this manuscript should be improved, with focus on reducing the redundancy and increasing coherence within a paragraph. I have listed several places that need such improvement, but please try to improve throughout the manuscript (not limited to my list). When these comments/suggestions are addressed in the manuscript, I recommend this manuscript to be published in GMD.

Major comments

1. I encourage the authors to clarify the following point carefully throughout the manuscript. In my understanding, the STOPS model seems to be a great modeling tool, mainly due to less computational burden. It might be particularly useful when it needs to explore several possibilities. However, I don't think STOPS itself improves any air quality prediction. Also, the authors already stated that STOPS simulation results are relatively similar to CMAQ. I think the significant improvement in simulated PM10 was contributed by constraining PM10 based on GOCI AOD, not by using the STOPS model. CMAQ with the constrained PM10 from GOCI-AOD should also simulate a more accurate Asian dust. In short, I think STOPS does not contribute to "more accurate" forecasting but could help for "quicker" forecasting. If the authors agree with me, please change any relevant parts throughout the manuscript.

2. I suggest adding more detailed information of STOPS in Section 2.1. It is not easy to picture what exactly the STOPS model does (why is it a hybrid Lagrangian-Eulerian model?). I found the short description on the abstract (line 21-23) and the Figure 1 in Czader et al. (2015) quite helpful, which could be added to Section 2.1. Please clarify model domain and dispersion process used in STOPS: 1) does STOPS accounts for vertical and horizontal dispersion as it transport, like FLEXPART, which

means it changes the number of grids carrying by STOPS over time?; 2) does STOPS carry a couple of grids in the defined STOPS domain or STOPS moves the defined STOPS domain over time (e.g., 61x61 gridcells in Section 4.1)?

3. I agree with the authors that the main reason for the PM10 underprediction in CMAQ is very likely missing dust emissions, as the threshold friction velocity calculation indicates. However, I don't agree with the authors on how to draw a conclusion that the model meteorology is accurate, mainly because the evaluation results, shown in Figure 3, are not comprehensive. Here are more specific questions related to the evaluation. First of all, why do the authors choose averaged values of 20 sites? I'd strongly prefer to see individual site evaluations. Alternatively, the individual site evaluation can be provided in supplementary material. Secondly, given that the long-range transport of Asian dust to influence South Korea, it is important to simulate correct meteorology from source regions to receptor regions. Would it be possible to include meteorological evaluations in Chinese source regions? Lastly, I encourage including more meteorological variables (such as precipitation, if there is any precipitation event during the event).

4. Please provide a brief description of the CMAQ dust emission parameterizations used in your forecast modeling. It will help readers to understand what the underpredicted threshold friction velocity affects to dust emissions.

Minor comments:

Title

I'd suggest changing a title. What about "Computationally efficient air quality forecasting tool: implementation of a hybrid Lagrangian-Eulerian model into CMAQ v5.0.2"?

Abstract

I'd strongly suggest re-writing this section. Overall abstract seems to sound quite redundant. Please consider taking the suggestions below.

[Figure]

Page 1; line 17-19 – Please consider moving this to the end of Abstract and either delete or modify this phrase ("for a more accurate prediction of Asian dust event in Korea"): see the major comment above. Page 1; line 20-21 – I'd suggest deleting sentence. It is mentioned in line 31-33. Page 1; line 24-27 – Please consider deleting this as well. Next a few sentences basically say the same information. Having this sentence, it sounds too redundant. Page 1; line 29-31 – I'd suggest modifying this. The following is my suggestion: "The underestimated PM10 concentration is very likely due to missing dust emissions in CMAQ rather than incorrectly simulated meteorology as the model meteorology agrees well with the observations." Page 1; line 32 – Please delete "we use the STOPS modeling system inside the CMAQ model, and", and please modify "we run several STOPS simulations using" to "we used the STOPS model with". Page 2; line 2-4 – Please shorten the sentence. "The simulated PM10 from the STOPS simulations were improved significantly and closely matched to surface observations". Page 2; line 5-9 – Please see my major comment 1.

Introduction

Page 2; line 18-21 - I'd suggest changing "Severe PM events . . . Gobi Desert" to "Dust emissions from Mongolia and Gobi Desert". Page 2; line 23 – please change "become" to "are". Page 2; line 29 – Please rephrase "the numerous factors such as meteorology and emissions ... PM concentrations". It sounds a bit unclear. Page 2; line 21 – Add "modeling" in front of "studies"; change "described" to "shown" and delete "simulation". Page 3; line 31 to Page 3; line 9 – This paragraph should be rewritten in order to deliver the key point clearly, which, I think, improving meteorology and emission inventory do not help better Asian dust forecasting due to the uncertainty in dust emission modeling. Besides, please delete the last sentence (Therefore, ~): the first part is too obvious to mention, and the second part is somewhat debatable (especially "primarily") and contradicts with "accurate meteorology" above. Page 3; line 25 – This "(STOPS, here-after)" should be moved above, where STOPS is mentioned in the first time. Page 3; line 22-35 – I found this paragraph This paragraph doesn't sound coherent. Please use

present tense to state goals and objectives and past tense for methods. Please also modify the paragraph based on my major comment 1. It is incorrect to say that STOPS enhance the PM predictions. Page 3; line 23 – Delete "simulated"; add "to" in front of "determine". Page 3; line 24 – Delete "particularly", as this study focuses on Asian dust event only.

2.2. Modeling system and experimental design

Page 5; line 4-5 – I think this sentence fits better in the end of next paragraph. Page 5; line 10 – why do you mean by "refer to the CAPPS emissions"? Page 5; line 18 – delete "for the simulation" Page 5; line 18-23 – Please shorten the sentences. Page 5; line 24 – Please remove "listed in Table 1" and list the date here.

2.3. In-situ and satellite measurements

Page 5; line 29 – "referred to" to "use" Page 5; line 36 – what is this "500 m resolution" for? Why is it different from AOD's 6 km resolution? Page 6; line 1 – "550 nm AOD" to "AOD at 550nm"

3.1 Comparison with surface measurement

Page 6; line 20-22 – Please define RMSE, IOA and MBE and explain what each measure indicates briefly. Page 6; line 26-29 – Please see the major comment 3. Page 6; line 30-36 – CMAQ dust emission modeling should be explained before this result. Please add the brief description in method section.

3.2 Comparison with satellite-based observation

Page 7; equations 4-6 – It looks like empirically derived method. Does the method by Roy et al. (2007) tested over the Korea as compared to more theoretical-based (Mie theory) optical properties? Is it reasonable to use it for Korea? Also, why isn't there no water uptake by organic aerosol [OM] in Eq 5? Figure 4 – It is good that the CMAQ AOD field shows removed areas with GOCI bad pixels. However, it would be also helpful to present CMAQ AOD without removing any areas in the supplementary

materials. It could show what GOCI might miss in those areas. Page 7; line 32 – delete "the same results" Page 7; line 34 – Do you actually mean "PM precursor" or "PM and its precursor"? If it is indeed specifically "PM precursor", please provide further explanation. Next sentence about meteorology should be re-considered (see major comment) Page 8; line 3 - please add year: Feb 22-24, 2015. Please make the same corrections throughout the manuscript, if possible.

Page 8; line 15-16 – please change "the high amounts of dust particles" to "the high dust concentrations". Page 8; line 19-20 – This should be modified with my major comment 1. I'd suggest changing to this: "We use STOPS to explore how to improve PM10 simulation."

4.2. PM10 forecasting using STOPS

Page 9; line 6-8 – This sentence is unnecessarily long. Please remove "that is, the . . . failed". Page 9; line 8-9 – This should be rephrased, esp. "the most recent and accurate input data". It makes me think about meteorology, emissions, initial and boundary conditions. If the constrained PM10 derived from GOCI AOD is only read in the first time, it is considered initial concentration and thus "input data". However, the way you used the constrained PM10 derived from GOCI AOD in Section 4.2.2 seems more than initialization and close to nudging. Page 9; line 13-18 – Please remove this part. This is out of placed and doesn't have much new information, in my opinion. If the authors want to make a point that the CMAQ with constrained PM using GOCI AOD is less desirable as a forecasting tool due to their long simulations, perhaps do it elsewhere (maybe the end of the paragraph). Page 9; line 18 – what do you mean by "dust core"? center of dust storm? Page 9; line 26- do you actually mean "on the STOPS domain"? Perhaps it is "on the STOPS results"? Also, perhaps "would be diminished" is better than "would be mitigated"?

4.2.1. Satellite-adjusted PM concentrations

This section is particularly confusing. Please re-write them and use figure or diagram

to help readers to understand the method.

Page 9; line 31 – Please remove "To provide $\sim$ AOD into account," and clarify "at the beginning of the updated forecast". Page 9; line 34 – Perhaps "as a constraint" is correct? Page 10 – Isn't the second paragraph better to move? Page 11; line 2-3 – Fix line break

4.2.2. Enhanced PM10 forecasting using STOPS

Page 11; line 22 - why did you said "were assumed to"? Page 11; line 29-30 – please shorten to "Figure 8 exhibits clear..." Page 11; line 27 – please add ", shown in Fig. 8," after using STOPS Page 11; line 32 – please change to "because of the poor dust emission modeling in CMAQ". Page 11; line 36$\sim$ - Isn't this already mentioned in Line 30? Page 12; line 32 – Remove "changed" in "To verify the changed horizontal"

Summary

Please revise the summary section if it is subject to the major comments.

Page 13; line 22 – "but with" to "but used" Page 13; line 24 – add comma between "dust events" and "we"

Tables & Figures

Table 2 – "Without Dust Events" to "Without dust events" Figure 1 – It is hard to find the site location. I was able to find only 17 sites. Can you use color symbol for sites? Figure 2 – It would be nice, if the dust event days were shown in the figure. Figure 6 – Does white space shown in the map represent for very low AOD or does it also include areas with missing pixels? Just in cases missing areas should be shown in white. Figure 7 – Please double check the caption. It says standard and constrained CMAQ runs, while "constrained CMAQ run" is never discussed in the main text.

Please also note the supplement to this comment:
http://www.geosci-model-dev-discuss.net/gmd-2016-180/gmd-2016-180-RC1-

supplement.pdf

---

## Referee Comment (RC2) · Anonymous Referee #2 · 6 Sep 2016

P3, line 24-35, grammatical errors. For describing what was done in this paper, the past tense would be used. Not just in this paragraph, many grammatical errors are in the text. Sentences are not conveying arguments smoothly that I need to read them a few times to understand authors' intention (such as P3, line 30-33 ). Sometimes, the wordings are redundant in carrying out the arguments (like p7, p9 line 5-10, p9, line 13-24). With the help of professional English editing and proof reading, the manuscript will be more concise and readable. P3, line 11, give citation (Byun and Schere, 2006) when the model is 1st mentioned in the paper. P3, line 27, "We utilized STOPS..." P3, line 29, "input data inside the modeling domain." P4, line 5, re-phase the sentence to

"A small sub-domain of STOPS was configured inside the CMAQ domain and it moves along with the mean wind from CMAQ." P4, line 9, the sentence is confusing, please re-write it. P4, line 10-11, "...is determined by the layer-averaged wind from the 1st model up to the top of planetary boundary layer (PBL), weighted by the layer thickness." P4, line 27, "but in this study, STOPS has been updated to v1.5 and implemented in CMAQ v5.0.2." P4, line 31-33, No need to give citation again for the CMAQ. "In this study, we configured the CMAQ model with a domain in a grid resolution of 27 km covering the northeastern part of Asia..." P4, line 29, the list and description of all the simulations – standard CMAQ, CMAQ with windblow dust, CMAQ with adjusted emission and four STOPS with adjusted emission are expected in the section titled as experimental design. It can be in its own section if appropriate. P5, line 1-2, "Gobi Desert which is a major source of Asian dust." P5, line 2, spell out full name of "CB05" and "AERO6" and provide citations. P5, line 5-22, missing CMAQ and WRF's model configuration. Please list physics options used in WRF and the schemes (such as advection, deposition, etc...) used in CMAQ. Also, the model configuration for STOPS should be described in this section. P5, line 24, please provide overview of the synoptic weather pattern during the dust event that will help readers to interpret the model result. P5, line 23-25, the paragraph should be re-written to give clear information about the simulation period and when the dust event happened. "The WRF-CMAQ simulations were conducted for the period of January 21st – February 28th, 2015 which included the first ten days for spin-up. Evaluations applied to the month of February, 2015 and the three-day Asian dust event occurred during February 22nd – 24th. The PM surface observations measured at the surface stations in Korea are listed in Table 1. P5, line 29, "This study used surface observational data..." P6, line 3, what does it mean for constraining of PM concentration? Is it through data assimilation? If so, it should be described in methodology section like 2.2. P6, line 30-35, what does the windblown dust module do in CMAQ? Any references for other studies using it? Figure 2 comparison shows almost no difference in PM predictions from simulations of standard CMAQ and CMAQ with dust module, even during the period of the dust event. If you lower the

threshold in the dust module, will the CMAQ be able to simulate the dust event? P7, line 4-20, I think it will be more appropriate to have these paragraphs in section 2.3 to describe how the satellite AOD used for CMAQ evaluations. Then, section 3.2 can focus on presenting the comparison and discussing the underestimation during the dust period. P8, section 4.1, it is out of place but better to be moved to section 2.2. P8, line 32, why the STOPS domain does not cover the whole Korean Peninsula? In this case, is the AQMS station at the east coast not included in the domain? P9, section 4.2, I cannot get the point of the section. Using half of the page, it repeats findings (CMAQ failed to simulation the dust event and STOPS could produce CMAQ's result with mush less computational time) that have already shown in the previous sections. This section should be re-written to be more concise and informative. P9, line 34, I cannot understand how can you add extra amount of PM directly to CMAQ without some kind of data assimilation technique? P10, Rather than improving the dust module in CMAQ, using satellite AOD to take into account the extra emission due to the dust event is one reasonable way to improve PM10 prediction for this study. But why the STOPS model is a tool for "a more accurate prediction" (as highlighted in the title)? STOPS is more efficient computationally than running the full CMAQ model? The improvement shown in STOPS results is due to the use of adjusted emission estimated according to the satellite data. By using the same adjust emission, can the CMAQ also produce better PM10 prediction compared to the standard CMAQ? P10, line 32, what is PMT? P11, line 8-16, the text talks about the CMAQ .vs. STOPS simulations but the figure is in CMAQ domains. And the caption indicates both are CMAQ simulations. Please clarify and use consistent names. P11, line 8, re-phase it to "Figure 7 shows the comparison of the PM10 concentration from CMAQ simulations using standard and adjusted emission". P11, line 33-37, I do not know what the "updated" is referring to. Use just "STOPS simulation" instead of "updated STOPS simulation" P12, line 6-7, re-phase to "the impact of the alternative emissions on the PM10 prediction highly depends on the durations of emission release and the impact was gone after the release ended." P12, line 17, '...AOD data contained missing data due to the cloud cover over the

study area . . ." P13, line 28-29, re-phase to "With reasonable meteorological input, the under-prediction of PM10 concentration was mainly due to the inaccurate estimation of dust emission during this period used in CMAQ." Figure 2, the CMAQ_dust simulation should be explained in the text and please briefly describe what is the dust module in CMAQ. Figure 7, caption: ". . .alternative emission estimated according to the GOCI-derived AOD."

---

## Author Comment (AC1) · 18 Sep 2016

**PLEASE NOTE**

Reviewers' questions are in standard text.

Manuscript text is in *italic*.

Personal communication for reviewer is in **bold**.

**Response to reviewer #1:**

This study by Jeon et al. implemented a new hybrid Lagrangian-Eulerian model, STOPS, into CMAQ, to improve the air quality forecasting. Jeon et al. use the STOPS modeling framework with constrained PM from geostationary satellite AOD to improve the Asian dust event that occurred in South Korea on Feb 22-24, 2015. It demonstrates well how STOPS could be useful in air quality forecast, particularly for the unusual air quality events such as Asian dust transport. The merit of using STOPS is on low computational burden compared to CMAQ, which can be critical for emergency forecasting. The manuscript is well within the scope of GMD. However, the manuscript requires some revisions. Please see my comments below. In addition to those comment, I believe science writing in this manuscript should be improved, with focus on reducing the redundancy and increasing coherence within a paragraph. I have listed several places that need such improvement, but please try to improve throughout the manuscript (not limited to my list). When these comments/suggestions are addressed in the manuscript, I recommend this manuscript to be published in GMD.

**The authors agreed with reviewer's comment about adding a detailed description of STOPS and in-line dust module in CMAQ v5.0.2, and additional meteorological evaluation results at each observational site. To that point, we have added a figure to briefly illustrate the basic concept of the STOPS model and also added an equation to better explain the in-line windblown dust module. Additionally, we have added the WRF evaluation results (statistics: RMSE, IOA and MBE) for all individual sites to depict a more comprehensive evaluation. Further, we shortened and revised the text so as to reduce redundancy, and have added comprehensive figures and tables for clarifying our results. Please see our responses to the specific comments.**

**Major Comments:**

1. I encourage the authors to clarify the following point carefully throughout the manuscript. In my understanding, the STOPS model seems to be a great modeling tool, mainly due to less computational burden. It might be particularly useful when it needs to explore several possibilities. However, I don't think STOPS itself improves any air quality prediction. Also, the authors already stated that STOPS simulation results are relatively similar to CMAQ. I think the significant improvement in simulated PM10 was contributed by constraining PM10 based on GOCI AOD, not by using the STOPS model. CMAQ with the constrained PM10 from GOCI-AOD should also simulate a more accurate Asian dust. In short, I think STOPS does not contribute to "more accurate" forecasting but could help for "quicker" forecasting. If the authors agree with me, please change any relevant parts throughout the manuscript.

**Thanks for the point. The authors agree that STOPS itself does not specifically improve any air quality prediction, but help for "quicker" forecasting. The significant improvement in the simulated $PM_{10}$ was contributed by constrained PM concentrations based on GOCI AOD. Thus, we revised all of the relevant parts and some sentences throughout the manuscript to avoid any possible misunderstanding from readers.**

2. I suggest adding more detailed information of STOPS in Section 2.1. It is not easy to picture what exactly the STOPS model does (why is it a hybrid Lagrangian-Eulerian model?). I found the short description on the abstract (line 21-23) and the Figure 1 in Czader et al. (2015) quite helpful, which could be added to Section 2.1. Please clarify model domain and dispersion process used in STOPS: 1) does STOPS accounts for vertical and horizontal dispersion as it transport, like FLEXPART, which means it changes the number of grids carrying by STOPS over time?; 2) does STOPS carry a couple of grids in the defined STOPS domain or STOPS moves the defined STOPS domain over time (e.g., 61x61 gridcells in Section 4.1)?

**As the reviewer suggested, we added a figure similar to Figure 1 in Czader et al., (2015) in the revised manuscript to show more detailed information of STOPS.**

*<Figure 1 in the revised manuscript>*

[Figure]

*Figure 1. Conceptual diagram showing the basic structure and movement of the STOPS domain inside the CMAQ domain.*

As shown in **Figure 1** in the revised manuscript, STOPS has sub-domain inside the CMAQ domain and it moves along with the mean wind in its domain. The vertical structure and the physical and chemical process in STOPS are exactly same as in the host CMAQ model except for the calculation of advection fluxes. CMAQ uses horizontal wind velocity (u and v) from WRF to calculate horizontal advection fluxes; while STOPS calculates the difference between a cell horizontal wind velocity and the mean horizontal velocity in STOPS domain, so it can consider the moving speed and direction of STOPS domain for the calculation of advection fluxes. Since the STOPS domain moves over time, the horizontal velocity from WRF should be adjusted based on the movement of STOPS domain. Although STOPS is Eulerian-Lagrangian based model, it is close to Eulerian model rather than Lagrangian. STOPS is almost similar to CMAQ but has small domain size. The reason why STOPS is much faster than full CMAQ is that the number of grid cells in STOPS domain is much smaller than those in CMAQ domain. We revised section 2.1 in the manuscript by adding a figure and a couple of sentences for the better explain of STOPS model.

*<Section 2.1 in the revised manuscript>*

*STOPS has the same vertical structure and simulates the same physical and chemical*

*processes as CMAQ, except for the calculation of advection fluxes. CMAQ uses horizontal wind velocity (u and v) from WRF to calculate horizontal advection fluxes, but STOPS calculates the difference between a cell horizontal wind velocity and the mean horizontal velocity in STOPS domain (Czader et al., 2015), so it can consider the moving speed and direction of STOPS domain for the calculation of advection fluxes. Since the STOPS domain moves over time, the horizontal velocity from WRF should be adjusted based on the movement of STOPS domain.*

3. I agree with the authors that the main reason for the PM10 underprediction in CMAQ is very likely missing dust emissions, as the threshold friction velocity calculation indicates. However, I don't agree with the authors on how to draw a conclusion that the model meteorology is accurate, mainly because the evaluation results, shown in Figure 3, are not comprehensive. Here are more specific questions related to the evaluation. First of all, why do the authors choose averaged values of 20 sites? I'd strongly prefer to see individual site evaluations. Alternatively, the individual site evaluation can be provided in supplementary material. Secondly, given that the long-range transport of Asian dust to influence South Korea, it is important to simulate correct meteorology from source regions to receptor regions. Would it be possible to include meteorological evaluations in Chinese source regions? Lastly, I encourage including more meteorological variables (such as precipitation, if there is any precipitation event during the event).

**Firstly, we added WRF evaluation results (RMSE, IOA and MBE) for 20 sites (S1-S20) in the revised supplementary document as the reviewer suggested. Table S3 in the revised supplementary document shows evenly high IOA and low biases at 20 individual sites, indicating that the simulated meteorology over Korea (receptor regions) is reasonably accurate.**

*<Table S3 in the revised supplementary document>*

**Table S3**. *Statistical parameters for the WRF simulation results during the entire simulation period (February 2015) at 20 observational sites. The location of each site is shown in Fig. 2 in the manuscript.*

| Sites | Temperature | Wind Speed |
|---|---|---|

| | RMSE | IOA | MBE | RMSE | IOA | MBE |
|---|---|---|---|---|---|---|
| S1 | 0.78 | 0.99 | -0.08 | 1.12 | 0.97 | 0.03 |
| S2 | 1.46 | 0.98 | 0.17 | 1.38 | 0.90 | 0.15 |
| S3 | 2.49 | 0.90 | -0.27 | 1.23 | 0.80 | -0.85 |
| S4 | 1.94 | 0.93 | 1.80 | 1.28 | 0.78 | -0.21 |
| S5 | 2.31 | 0.93 | 1.48 | 1.13 | 0.84 | -0.40 |
| S6 | 2.31 | 0.93 | 1.04 | 1.89 | 0.91 | 1.49 |
| S7 | 2.48 | 0.96 | -1.46 | 1.96 | 0.77 | 1.43 |
| S8 | 2.58 | 0.93 | -1.58 | 1.61 | 0.87 | 1.25 |
| S9 | 1.40 | 0.94 | 1.39 | 1.19 | 0.86 | 1.12 |
| S10 | 1.42 | 0.95 | 1.41 | 1.87 | 0.91 | 1.21 |
| S11 | 2.02 | 0.97 | -1.06 | 2.03 | 0.75 | 1.45 |
| S12 | 2.70 | 0.78 | -2.35 | 1.34 | 0.92 | 0.94 |
| S13 | 2.11 | 0.94 | 1.24 | 1.24 | 0.88 | 0.85 |
| S14 | 1.59 | 0.95 | 1.01 | 2.07 | 0.93 | 1.46 |
| S15 | 2.67 | 0.89 | -2.29 | 2.37 | 0.76 | 1.90 |
| S16 | 1.39 | 0.98 | 0.43 | 1.59 | 0.89 | 0.90 |
| S17 | 2.48 | 0.84 | -1.71 | 1.98 | 0.74 | 1.36 |
| S18 | 1.60 | 0.96 | -1.09 | 2.64 | 0.72 | 1.27 |
| S19 | 1.58 | 0.95 | 1.17 | 2.03 | 0.82 | 1.02 |
| S20 | 1.12 | 0.96 | 0.98 | 1.59 | 0.89 | 0.90 |
| Average | 1.92 | 0.93 | 0.01 | 1.68 | 0.85 | 0.86 |

Secondly, the authors agree that the meteorology in source regions (China and Mongolia) is also important for the simulation of long-range transport of Asian dust; hence the data need to be evaluated. This study showed accurate meteorology only in receptor regions (Korea) not in source regions (China and Mongolia) due to the limited availability of the data. As the reviewer indicated, uncertainty in meteorology (particularly in source regions) could be one of possible reason for the $PM_{10}$ underestimation. We have added the requisite description regarding the uncertainty in meteorology in section 3.1 in the revised manuscript.

*<Section 3.1 in the revised manuscript>*

*As shown in Fig. 4 and Table S3, meteorological fields such as temperature and wind speed over receptor regions (Korea) showed close agreement with observations, even during the Asian dust period. It suggests that the underestimated $PM_{10}$ concentration was likely due to the uncertainty in meteorology over source regions (China and Mongolia), and/or faulty*

*estimation of dust emissions for the CMAQ simulation. We attributed the main reason for the PM$_{10}$ underestimation to poorly estimated dust emission because CMAQ showed poor performance only during the Asian dust event days.*

**Finally, there were several challenges in obtaining observational data in China and Mongolia. Also, the surface data for this study provided only temperature and wind variability. For these reasons we could not include meteorological evaluations in source regions and evaluation results for other factors such as precipitation.**

4. Please provide a brief description of the CMAQ dust emission parameterizations used in your forecast modeling. It will help readers to understand what the underpredicted threshold friction velocity affects to dust emissions.

**As the reviewer suggested, we provided a brief description of the CMAQ in-line windblown dust module as shown below.**

*<Section 3.1 in the revised manuscript>*

*The module calculates the vertical dust emission flux (F) by following formula described by Fu et al. (2014).*

$$F = \sum_{i=1}^{M} \sum_{j=1}^{N} K \times A \times \frac{\rho}{g} \times S_i \times SEP \times u_* \times (u_*^2 - u_{*ti,j}^2)$$

*where i and j represent the type of erodible land and soil, K is the ratio between vertical and horizontal flux, A is the particle supply limitation, $\rho$ is the air density, g is the gravitational constant, $S_i$ is the area of the dust source, SEP is the soil erodible potential, $u_*$ is the friction velocity, and $u_{*ti,j}$ denotes the threshold friction velocity.*

**Also, we added a sentence describing the importance of threshold friction velocity on the calculation of dust emission flux to better explain the reason for the underestimated dust emission from the CMAQ in-line module.**

*<Section 3.1 in the revised manuscript>*

*Several studies (e.g. Choi et al., 2008; Fu et al., 2014) have reported that the threshold friction velocity plays a key role in the calculation of dust emission flux because the threshold*

*can determine the possibility of the lifting of dust particles.*

**Minor comments:**

1. <Title>: I'd suggest changing a title. What about "Computationally efficient air quality forecasting tool: implementation of a hybrid Lagrangian-Eulerian model into CMAQ v5.0.2"?

**The authors agreed to change the title as the reviewer suggested. However, we added a phrase "for a prediction of Asian dust" to emphasize that this is a case study for an Asian dust event. Also, we used "STOPS model" instead of "a hybrid Lagrangian-Eulerian model", because we thought "a hybrid Lagrangian-Eulerian model" is too generic to be used in the title. In conclusion, we changed the title of this study as "Computationally efficient air quality forecasting tool: implementation of STOPS model into CMAQ v5.0.2 for a prediction of Asian dust".**

2. <Abstract> : I'd strongly suggest re-writing this section. Overall abstract seems to sound quite redundant. Please consider taking the suggestions below.

Page 1; line 17-19 – Please consider moving this to the end of Abstract and either delete or modify this phrase ("for a more accurate prediction of Asian dust event in Korea"): see the major comment above.

Page 1; line 20-21 – I'd suggest deleting sentence. It is mentioned in line 31-33.

Page 1; line 24-27 – Please consider deleting this as well. Next a few sentences basically say the same information. Having this sentence, it sounds too redundant.

Page 1; line 29-31 – I'd suggest modifying this. The following is my suggestion: "The underestimated PM10 concentration is very likely due to missing dust emissions in CMAQ rather than incorrectly simulated meteorology as the model meteorology agrees well with the observations."

Page 1; line 32 – Please delete "we use the STOPS modeling system inside the CMAQ model, and", and please modify "we run several STOPS simulations using" to "we used the STOPS model with".

Page 2; line 2-4 – Please shorten the sentence. "The simulated PM10 from the STOPS simulations were improved significantly and closely matched to surface observations".

Page 2; line 5-9 – Please see my major comment 1.

**We re-wrote the Abstract section based on the reviewer's comments. We shortened and changed the sentences as the reviewer suggested and deleted unnecessary sentences to reduce the redundancy.**

*<Abstract in the revised manuscript>*

***Abstract.*** *This study suggests a new modeling framework using a hybrid Lagrangian-Eulerian based modeling tool (the Screening Trajectory Ozone Prediction System, STOPS) for a prediction of an Asian dust event in Korea. The new version of STOPS (v1.5) has been implemented into the Community Multi-scale Air Quality (CMAQ) model version 5.0.2. The STOPS modeling system is a moving nest (Lagrangian approach) between the source and the receptor inside the host Eulerian CMAQ model. The proposed model generates simulation results that are relatively consistent with those of CMAQ but within a comparatively shorter computational time period. We find that standard CMAQ generally underestimates $PM_{10}$ concentrations during the simulation period (February 2015) and fails to capture $PM_{10}$ peaks during Asian dust events (22-24 February, 2015), The underestimated $PM_{10}$ concentration is very likely due to missing dust emissions in CMAQ rather than incorrectly simulated meteorology as the model meteorology agrees well with the observations. To improve the underestimated $PM_{10}$ results from CMAQ, we used the STOPS model with constrained PM concentrations based on aerosol optical depth (AOD) data from Geostationary Ocean Color Imager (GOCI), reflecting real-time initial and boundary conditions of dust particles near the Korean Peninsula. The simulated $PM_{10}$ from the STOPS simulations were improved significantly and closely matched to surface observations. With additional verification of the capabilities of the methodology on concentration estimations and more STOPS simulations for various time periods, STOPS model could prove to be a useful tool not just for the predictions of Asian dust but also for other unexpected events such as wildfires and upset emissions events.*

3. <1. Introduction>

Page 2; line 18-21 - I'd suggest changing "Severe PM events … Gobi Desert" to "Dust emissions from Mongolia and Gobi Desert".

Page 2; line 23 – please change "become" to "are".

Page 2; line 29 – Please rephrase "the numerous factors such as meteorology and emissions ... PM concentrations". It sounds a bit unclear.

Page 2; line 21 – Add "modeling" in front of "studies"; change "described" to "shown" and delete "simulation".

Page 3; line 31 to Page 3; line 9 – This paragraph should be rewritten in order to deliver the key point clearly, which, I think, improving meteorology and emission inventory do not help better Asian dust forecasting due to the uncertainty in dust emission modeling. Besides, please delete the last sentence (Therefore, ~): the first part is too obvious to mention, and the second part is somewhat debatable (especially "primarily") and contradicts with "accurate meteorology" above.

Page 3; line 25 – This "(STOPS, hereafter)" should be moved above, where STOPS is mentioned in the first time.

Page 3; line 22-35 – I found this paragraph This paragraph doesn't sound coherent. Please use present tense to state goals and objectives and past tense for methods. Please also modify the paragraph based on my major comment 1. It is incorrect to say that STOPS enhance the PM predictions.

Page 3; line 23 – Delete "simulated"; add "to" in front of "determine".

Page 3; line 24 – Delete "particularly", as this study focuses on Asian dust event only.

**We revised the Introduction section based on the reviewer's comments. We rewrote some sentences more clearly and removed a couple of unnecessary sentences as the reviewer suggested.**

4. <2.2 Modeling system and experimental design>

Page 5; line 4-5 – I think this sentence fits better in the end of next paragraph.

Page 5; line 10 – why do you mean by "refer to the CAPPS emissions"?

Page 5; line 18 – delete "for the simulation"

Page 5; line 18-23 – Please shorten the sentences.

Page 5; line 24 – Please remove "listed in Table 1" and list the date here.

**We shortened, moved and deleted some sentences in section 2.2 (in the revised manuscript) as the reviewer suggested.**

5. <2.3 In-situ and satellite measurements>

Page 5; line 29 – "referred to" to "use"

Page 5; line 36 – what is this "500 m resolution" for? Why is it different from AOD's 6 km resolution?

Page 6; line 1 – "550 nm AOD" to "AOD at 550nm"

**500 m and 6 km are the resolutions of original GOCI data and retrieved one by Choi et al. (2016) algorithm. The retrieved GOCI data with a 6 km resolution were used in this study. We corrected two phrases in section 2.3 (in the revised manuscript) as the reviewer suggested.**

6. <3.1 Comparison with surface measurement>

Page 6; line 20-22 – Please define RMSE, IOA and MBE and explain what each measure indicates briefly.

Page 6; line 26-29 – Please see the major comment 3.

Page 6; line 30-36 – CMAQ dust emission modeling should be explained before this result. Please add the brief description in method section.

**As the reviewer suggested, we added brief description of the statistical parameters used in this study (RMSE, IOA and MBE) in section 2.3 (in the revised manuscript).**

*<Section 2.3 in the revised manuscript>*

*The following statistical parameters were used for the evaluation of the performance of WRF and CMAQ simulations: Index Of Agreement (IOA), Mean Bias Error (MBE) and Root Mean Square Error (RMSE). These are defined as:*

$$IOA = 1 - \frac{\sum_{i=1}^{N}(P_i - O_i)^2}{\sum_{i=1}^{N}(|P_i - \bar{P}| + |O_i - \bar{O}|)^2}$$

$$MBE = \frac{\sum_{i=1}^{N}(P_i - O_i)}{N}$$

$$RMSE = \sqrt{\frac{\sum_{i=1}^{N}(P_i - O_i)^2}{N}}$$

*where N is number of data points and $P_i$ and $O_i$ denote CMAQ-simulated and observed concentrations, respectively.*

**We also added the WRF evaluation results at individual sites in Table S3 (in the revised supplementary document), and a formula for dust estimation used in CMAQ in-line dust module in section 3.1 (in the revised manuscript) for the better explanation. Please see our responses for question 3 and 4.**

7. <3.2 Comparison with satellite-based observation>

Page 7; equations 4-6 – It looks like empirically derived method. Does the method by Roy et al. (2007) tested over the Korea as compared to more theoretical-based (Mie theory) optical properties? Is it reasonable to use it for Korea? Also, why isn't there no water uptake by organic aerosol [OM] in Eq 5?

Figure 4 – It is good that the CMAQ AOD field shows removed areas with GOCI bad pixels. However, it would be also helpful to present CMAQ AOD without removing any areas in the supplementary materials. It could show what GOCI might miss in those areas.

Page 7; line 32 – delete "the same results"

Page 7; line 34 – Do you actually mean "PM precursor" or "PM and its precursor"? If it is indeed specifically "PM precursor", please provide further explanation. Next sentence about meteorology should be re-considered (see major comment)

Page 8; line 3 - please add year: Feb 22-24, 2015. Please make the same corrections throughout the manuscript, if possible.

Page 8; line 15-16 – please change "the high amounts of dust particles" to "the high dust concentrations".

Page 8; line 19-20 – This should be modified with my major comment 1. I'd suggest changing to this: "We use STOPS to explore how to improve PM10 simulation."

Firstly, the empirical method used in this study has successfully been tested in East Asia (Park et al., 2011; Song et al., 2008) which is preferred to the Mie theory in this region. This is mainly because of the fact that aerosols properties including size distribution have not been precisely characterized in this region to allow us to use the Mie-theory extinction coefficient calculations. This issue was partly discussed in two mentioned papers. The OM hygroscpocitiy are highly uncertain, and to best of our knowledge it has not been parameterized yet. It should be mentioned that the significant portion of dust particles is $NH_4NO_3$ and $SO_4^{2-}$, therefore OM concentrations are not strongly prominent.

Secondly, as the reviewer suggested, we made a figure showing the CMAQ AOD without masking of GOCI bad pixels (Please see the figure shown below). However, it does not entirely match with previous figure (Figure 5-(b) in the revised manuscript) because bad pixels in GOCI were not filtered out for the calculation of monthly mean AOD from CMAQ. Although the below figure shows the CMAQ-derived AOD over whole areas in the modeling domain, it cannot be directly compared with figures in Figure 5 (in the revised manuscript). For this reason, the authors decided not to add the below figure to the supplementary document to avoid unnecessary argument.

[Figure]

Lastly, we revised all the sentences in section 3.2 as the reviewer suggested.

8. <4.2 PM$_{10}$ forecasting using STOPS>

Page 9; line 6-8 – This sentence is unnecessarily long. Please remove "that is, the … failed".

Page 9; line 8-9 – This should be rephrased, esp. "the most recent and accurate input data". It makes me think about meteorology, emissions, initial and boundary conditions. If the constrained PM10 derived from GOCI AOD is only read in the first time, it is considered initial concentration and thus "input data". However, the way you used the constrained PM10 derived from GOCI AOD in Section 4.2.2 seems more than initialization and close to nudging.

Page 9; line 13-18 – Please remove this part. This is out of placed and doesn't have much new information, in my opinion. If the authors want to make a point that the CMAQ with constrained PM using GOCI AOD is less desirable as a forecasting tool due to their long simulations, perhaps do it elsewhere (maybe the end of the paragraph).

Page 9; line 18 – what do you mean by "dust core"? center of dust storm?

Page 9; line 26- do you actually mean "on the STOPS domain"? Perhaps it is "on the STOPS results"? Also, perhaps "would be diminished" is better than "would be mitigated"?

**We removed and revised a couple of unnecessary sentences and confusing phrases as the reviewer suggested (Section 4 in the revised manuscript).**

9. <4.2.1 Satellite-adjusted PM concentrations>: This section is particularly confusing. Please re-write them and use figure or diagram to help readers to understand the method.

Page 9; line 31 – Please remove "To provide ~ AOD into account," and clarify "at the beginning of the updated forecast".

Page 9; line 34 – Perhaps "as a constraint" is correct?

Page 10 – Isn't the second paragraph better to move?

Page 11; line 2-3 – Fix line break 4.2.2.

**As the reviewer suggested, we re-wrote section 4.1 (in the revised manuscript) to better explain the method we used for PM constraining. We added a figure, which briefly describes entire procedures of the new PM forecasting using STOPS with GOCI-derived AOD data.**

*<Figure S3 in the revised supplementary document>*

[Figure]

**Figure S3**. *Schematic flowchart describing the procedures of the new PM forecasting by STOPS using the real-time AOD data from GOCI.*

10. <4.2.2 Enhanced PM10 forecasting using STOPS>

Page 11; line 22 - why did you said "were assumed to"?

Page 11; line 29-30 – please shorten to "Figure 8 exhibits clear…"

Page 11; line 27 – please add ", shown in Fig. 8," after using STOPS

Page 11; line 32 – please change to "because of the poor dust emission modeling in CMAQ".

Page 11; line 36~ - Isn't this already mentioned in Line 30?

Page 12; line 32 – Remove "changed" in "To verify the changed horizontal"

**We removed some unnecessary sentences, and revised all of the addressed phrases and sentences as the reviewer suggested in order to reduce redundancy (Section 4.2 in the revised manuscript).**

11. <Summary>: Please revise the summary section if it is subject to the major comments.

Page 13; line 22 – "but with" to "but used"

Page 13; line 24 – add comma between "dust events" and "we"

We revised the Summary by considering all of the changes in each section in the revised manuscript.

12. <Table & Figures>

Table 2 – "Without Dust Events" to "Without dust events"

Figure 1 – It is hard to find the site location. I was able to find only 17 sites. Can you use color symbol for sites?

Figure 2 – It would be nice, if the dust event days were shown in the figure.

Figure 6 – Does white space shown in the map represent for very low AOD or does it also include areas with missing pixels? Just in cases missing areas should be shown in white.

Figure 7 – Please double check the caption. It says standard and constrained CMAQ runs, while "constrained CMAQ run" is never discussed in the main text.

**We corrected "Without Dust Events" in Table 2 (in the revised manuscript) to "Without dust events", changed Figure 2 (in the revised manuscript) by adding 3 missing sites and using color symbols, and marked the Asian dust event days in Figure 3 (in the revised manuscript). Also, we revised caption in Figure 7 and 8 (in the revised manuscript) for the better explanation.**

*<Figure 2, 3, 7 and 8 in the revised manuscript>*

[Figure]

***Figure 2.*** *Domains for the WRF, CMAQ and STOPS modeling. The right panel shows the location of the air quality monitoring stations (AQMS) and automatic weather system (AWS) sites used in this study.*

[Figure]

*Figure 3. Time series of observed (OBS, blue dots) and simulated (CMAQ: red line, CMAQ_Dust: black dashed line) PM₁₀ concentrations in February 2015. The values are averaged values for 20 AQMS sites: CMAQ_Dust is closely coupled with the standard CMAQ modeling results (red line).*

[Figure]

*Figure 7. The GOCI-derived AOD (550 nm) from 13:30 LST on 21 February to 12:30 LST on 22 February in 2015. The white-colored areas represent missing pixels.*

[Figure]

***Figure* 8.** *Difference of the simulated PM$_{10}$ concentrations ($\mu g$ m$^{-3}$) between the standard CMAQ run (left) and STOPS forecasting run with alternative emission estimated according to GOCI-derived AOD (right) inside the STOPS domain at 12:00 LST on 22 February in 2015.*

---

## Author Comment (AC2) · 18 Sep 2016

**PLEASE NOTE**

Reviewers' questions are in standard text.

Manuscript text is in *italic*.

Personal communication for reviewer is in **bold**.

**Response to reviewer #2:**

**The authors agreed with reviewer's comment regarding the necessity of the model configurations for WRF and CMAQ simulations, synoptic weather chart in the Asian dust event day, detailed description of in-line dust module in CMAQ v5.0.2, and more clear explanation of the methodology we used for STOPS forecasting. We added a couple of figures and tables, and additional description for them for better understanding, and revised a lot of sentences based on the reviewer's suggestion to reduce redundancy. Also, we revised a couple of confusing and misleading paragraphs in the manuscript with the professional English editing and proof reading to make the manuscript more concise and readable.**

**Again, the authors responded to most of the reviewer's comments and strengthened our revised manuscript and supplementary document. Please see our responses to the specific comments.**

**Specific Comments:**

1. P3, line 24-35, grammatical errors. For describing what was done in this paper, the past tense would be used. Not just in this paragraph, many grammatical errors are in the text. Sentences are not conveying arguments smoothly that I need to read them a few times to understand authors' intention (such as P3, line 30-33 ). Sometimes, the wordings are redundant in carrying out the arguments (like p7, p9 line 5-10, p9, line 13-24). With the help of professional English editing and proof reading, the manuscript will be more concise and readable.

**The authors revised all of the confusing and misleading paragraphs throughout the manuscript with the professional English editing and proof reading to make the**

**manuscript more concise and readable.**

2. P3, line 11, give citation (Byun and Schere, 2006) when the model is 1st mentioned in the paper.

   **We added a citation, "Byun and Schere, 2006", in the sentence.**

3. P3, line 27, "We utilized STOPS: : :",
   P3, line 29, "input data inside the modeling domain."

   **We corrected the sentences as suggested by the reviewer.**

4. P4, line 5, re-phase the sentence to C1 "A small sub-domain of STOPS was configured inside the CMAQ domain and it moves along with the mean wind from CMAQ."

   **We revised the sentence as the reviewer suggested, and added a figure in the revised manuscript for the better understanding from readers.**

*<Figure 1 in the revised manuscript>*

[Figure]

*Figure 1. Conceptual diagram showing the basic structure and movement of the STOPS domain inside the CMAQ domain.*

5. P4, line 9, the sentence is confusing, please rewrite it.

**We re-wrote the sentence to clearly explain how STOPS calculates advection fluxes.**

*<Section 2.1 in the revised manuscript>*

*STOPS has the same vertical structure and simulates the same physical and chemical processes as CMAQ, except for the calculation of advection fluxes. CMAQ uses horizontal wind velocity (u and v) from WRF to calculate horizontal advection fluxes, but STOPS calculates the difference between a cell horizontal wind velocity and the mean horizontal velocity in STOPS domain (Czader et al., 2015), so it can consider the moving speed and direction of STOPS domain for the calculation of advection fluxes. Since the STOPS domain moves over time, the horizontal velocity from WRF should be adjusted based on the movement of STOPS domain.*

6. P4, line 10-11, ": : :is determined by the layer-averaged wind from the 1st model up to the top of planetary boundary layer (PBL), weighted by the layer thickness.",

P4, line 27, "but in this study, STOPS has been updated to v1.5 and implemented in CMAQ v5.0.2.",

P4, line 31-33, No need to give citation again for the CMAQ. "In this study, we configured the CMAQ model with a domain in a grid resolution of 27 km covering the northeastern part of Asia: : :"

**We revised the sentences as the reviewer suggested.**

7. P4, line 29, the list and description of all the simulations – standard CMAQ, CMAQ with windblow dust, CMAQ with adjusted emission and four STOPS with adjusted emission are expected in the section titled as experimental design. It can be in its own section if appropriate.

**We changed the title of section 2.2 from "*2.2. Modeling system and experimental***

*design*" to "*2.2. Modeling system*" because the section does not include any experimental procedure. We have included the descriptions of each simulation (CMAQ and STOPS) in their relevant sections to better explain the methodology, data and options used for each simulation case.

8. P5, line 1-2, "Gobi Desert which is a major source of Asian dust."

**We corrected the sentence as suggested by the reviewer.**

9. P5, line 2, spell out full name of "CB05" and "AERO6" and provide citations.

**We added the full names and citations for them in the revised manuscript.**

*<Section 2.2 in the revised manuscript>*

*The Carbon Bond chemical mechanism (CB05) (Yarwood et al., 2005) and the AERO6 aerosol module (Nolte et al., 2015) were used for gas-phase and aerosol chemical mechanisms, and initial and boundary conditions were obtained from the standard CMAQ profile.*

10. P5, line 5-22, missing CMAQ and WRF's model configuration. Please list physics options used in WRF and the schemes (such as advection, deposition, etc: : :) used in CMAQ. Also, the model configuration for STOPS should be described in this section.

**We added model configurations for WRF and CMAQ simulations in the revised supplementary document. Also, we moved section 4.1 (Configuration of STOPS) to this section as the reviewer suggested.**

*<Table S1 and S2 in the revised supplementary document>*

***Table S1***. *Configuration and detailed physical options for WRF simulation*

| Number of grids | $181 \times 143$ |
|---|---|

| | |
|---|---|
| Horizontal resolution | 27 km |
| Vertical layers | 33 layers |
| Initial data | 1°× 1° NCEP Final Operational Global Analysis (FNL) |
| Microphysics option | WSM 3-class simple ice scheme |
| Radiation option | RRTM (long wave) / Dudhia (short wave) scheme |
| Surface layer option | Monin-Obukhov (Janic Eta) scheme |
| Land-surface option | Unified Noah land-surface model |
| PBL option | YSU scheme |
| Cumulus option | Kain-Fritsch (new Eta) scheme |

**Table S2**. Same as Table S1, but for CMAQ

| | |
|---|---|
| Meteorology | WRF |
| Number of grids | 174 × 128 |
| Horizontal resolution | 27 km |
| Vertical layers | 15 layers |
| Chemical mechanism | CB05 (gas-phase) / AERO6 (aerosol) |
| Chemical solver | Smvgear |
| Horizontal advection | Yamo |
| Horizontal diffusion | Multiscale |
| Vertical advection | WRF |
| Vertical diffusion | ACM2 |
| Deposition | M3dry |
| Anthropogenic emissions | MIX-2010 / CAPSS 2011 |
| Dust emission model | In-line windblown dust model |

11. P5, line 24, please provide overview of the synoptic weather pattern during the dust event that will help readers to interpret the model result.

**We added two synoptic weather charts in the revised supplementary document to show the synoptic weather pattern on the first day of the Asian dust event (22 February, 2015), which resulted in the transport of massive dust from Mongolia region to the Korean Peninsula.**

*<Figure S1 in the revised supplementary document>*

[Figure]

**Figure S1**. *Surface-level synoptic weather chart near the Korean Peninsula on 22 February in 2015, which is the first day of the Asian dust event in this study.*

*<Section 2.2 in the revised manuscript>*

*During the event days, massive dust over the GOBI desert and Mongolia region was transported to the Korean Peninsula. This happened due to the southeastward wind resulting from high pressure over the Mongolia region and low pressure over the northeastern part of China (Fig. S1 in the supplementary document).*

12. P5, line 23-25, the paragraph should be re-written to give clear information about the simulation period and when the dust event happened. "The WRF-CMAQ simulations were conducted for the period of January 21st – February 28th, 2015 which included the first ten days for spin-up. Evaluations applied to the month of February, 2015 and the three-day Asian dust event occurred during February 22nd – 24th. The PM surface observations measured at

the surface stations in Korea are listed in Table 1.

**We re-wrote the paragraph as the reviewer's suggested.**

13. P5, line 29, "This study used surface observational data: : :"

**We revised the sentence as suggested by the reviewer.**

14. P6, line 3, what does it mean for constraining of PM concentration? Is it through data assimilation? If so, it should be described in methodology section like 2.2.

**We did not use data assimilation technique for constraining PM concentration in forecasting. Usually, the data assimilation techniques are computationally more expensive than the simplified constraining approach. As described in section 4.1 in the revised manuscript, we regarded the GOCI-derived AOD as a surrogate for PM emissions and hence indirectly constrained the original PM concentrations by using the alternative emissions. The GOCI-derived AOD was converted to emission unit and the converted emission values were used for the STOPS forecasting. Section 4.1 in the revised manuscript contains more detailed description for the method we used for STOPS forecasting with GOCI-derived AOD.**

15. P6, line 30-35, what does the windblown dust module do in CMAQ? Any references for other studies using it? Figure 2 comparison shows almost no difference in PM predictions from simulations of standard CMAQ and CMAQ with dust module, even during the period of the dust event. If you lower the C2 threshold in the dust module, will the CMAQ be able to simulate the dust event?

**We provided a brief description of the CMAQ in-line windblown dust module and a reference for it in section 3.1 in the revised manuscript.**

*<Section 3.1 in the revised manuscript>*

*The module calculates the vertical dust emission flux (F) by following formula described by*

*Fu et al. (2014).*

$$F = \sum_{i=1}^{M} \sum_{j=1}^{N} K \times A \times \frac{\rho}{g} \times S_i \times SEP \times u_* \times \left(u_*^2 - u_{*ti,j}^2\right)$$

*where i and j represent the type of erodible land and soil, K is the ratio between vertical and horizontal flux, A is the particle supply limitation, $\rho$ is the air density, g is the gravitational constant, $S_i$ is the area of the dust source, SEP is the soil erodible potential, $u_*$ is the friction velocity, and $u_{*ti,j}$ denotes the threshold friction velocity.*

**When we used the threshold values suggested by Fu et al. (2014), which are lower than standard ones, the simulated PM$_{10}$ concentrations over China, particularly in areas adjacent to the Gobi Desert and its downwind side increased as demonstrated by Fu et al. (2014). But the increase in Korea was relatively minimal and the result did not show reasonable agreement with observation. In this study, the average value of the simulated two-meter temperature during the period was 274.87 K, which was significantly lower than that founded by Fu et al. (2014) (286.30 K). The low friction velocity values below the threshold came from the cold weather conditions over the East Asia during the simulation period. We concluded that the employment of the in-line windblown dust module in CMAQ simulations did not provide discernible enhancement in PM$_{10}$ concentrations because of lower friction velocity than the threshold in the module. These are the reason why we thought a new modeling frame work for the prediction of Asian dust event.**

16. P7, line 4-20, I think it will be more appropriate to have these paragraphs in section 2.3 to describe how the satellite AOD used for CMAQ evaluations. Then, section 3.2 can focus on presenting the comparison and discussing the underestimation during the dust period.

**As suggested the reviewer, we moved the paragraphs in section 3.2 to section 2.3 in the revised manuscript, so section 3.2 can focus on presenting the comparison and discussing the underestimation during the dust period.**

17. P8, section 4.1, it is out of place but better to be moved to section 2.2.

**We moved section 4.1 (in the previous manuscript) to section 2.2 (in the revised manuscript) as the reviewer suggested.**

18. P8, line 32, why the STOPS domain does not cover the whole Korean Peninsula? In this case, is the AQMS station at the east coast not included in the domain?

**We found a problem with the initial position of the STOPS domain. The location of domain center was not 40˚ N, 119˚ E, but 40˚ N, 121˚ E, so we corrected relevant parts in the revised manuscript (Figure 2 and 10).**

19. P9, section 4.2, I cannot get the point of the section. Using half of the page, it repeats findings (CMAQ failed to simulation the dust event and STOPS could produce CMAQ's result with mush less computational time) that have already shown in the previous sections. This section should be re-written to be more concise and informative.

**We re-wrote section 4 in the revised manuscript to better explain the method we used for a new PM forecasting using STOPS. We added a figure, which briefly describes entire procedures of the PM forecasting using STOPS with GOCI-derived AOD data.**

*<Figure S3 in the revised supplementary document>*

[Figure]

*Figure S3*. *Schematic flowchart describing the procedure of the new PM forecasting by STOPS using the real-time AOD data from GOCI.*

20. P9, line 34, I cannot understand how can you add extra amount of PM directly to CMAQ without some kind of data assimilation technique?

**Please see our response for question 14.**

21. P10, Rather than improving the dust module in CMAQ, using satellite AOD to take into account the extra emission due to the dust event is one reasonable way to improve PM10 prediction for this study. But why the STOPS model is a tool for "a more accurate prediction" (as highlighted in the title)? STOPS is more efficient computationally than running the full CMAQ model? The improvement shown in STOPS results is due to the use of adjusted emission estimated according to the satellite data. By using the same adjust emission, can the CMAQ also produce better PM10 prediction compared to the standard CMAQ?

**As the reviewer addressed, the significant improvement in the simulated PM$_{10}$ was contributed by constrained PM concentrations based on GOCI AOD. Even though we used CMAQ instead of STOPS, it would produce the similar results as in STOPS. However, this study assumes a real forecasting situation. In the case of the massive dust transport is captured by satellite measurement, the current forecasting results should be**

replaced in a very short time period before the dust storm reaches the receptor regions (Korea in this study). A new forecasting using CMAQ with GOCI AOD cannot be done within a few minutes. Thus, the computational efficiency of STOPS is the most important benefit, which allows the near real-time update of PM forecasting results.

As the reviewer suggested, we revised a couple of misleading sentences throughout the manuscript by saying that STOPS itself does not improve any air quality prediction, but help for "quicker" forecasting.

22. P10, line 32, what is PMT?

The PMT is the same as $PMT_{i,j}$, the estimated emission rate of total PM in each grid cell. We changed PMT to $PMT_{i,j}$ for the better understanding from readers.

23. P11, line 8-16, the text talks about the CMAQ .vs. STOPS simulations but the figure is in CMAQ domains. And the caption indicates both are CMAQ simulations. Please clarify and use consistent names.

We changed Figure 8 in the revised manuscript to show PM$_{10}$ concentrations inside the STOPS domain, and revised its caption for clear description.

*<Figure 8 in the revised manuscript>*

[Figure]

**Figure 8.** *Difference of the simulated PM$_{10}$ concentrations ($\mu g$ $m^{-3}$) between the standard CMAQ run (left) and STOPS forecasting run with alternative emission estimated according to*

*GOCI-derived AOD (right) inside the STOPS domain at 12:00 LST on 22 February in 2015.*

24. P11, line 8, re-phase it to "Figure 7 shows the comparison of the PM10 concentration from CMAQ simulations using standard and adjusted emission".

P11, line 33-37, I do not know what the "updated" is referring to. Use just "STOPS simulation" instead of "updated STOPS simulation"

P12, line 6-7, re-phase to "the impact of the alternative emissions on the PM10 prediction highly depends on the durations of emission release and the impact was gone after the release ended."

P12, line 17, ': : :AOD data contained missing data due to the cloud cover over the C3 study area : : :"

P13, line 28-29, re-phase to "With reasonable meteorological input, the under-prediction of PM10 concentration was mainly due to the inaccurate estimation of dust emission during this period used in CMAQ."

**Thanks. We revised the sentences as the reviewer suggested.**

25. Figure 2, the CMAQ_dust simulation should be explained in the text and please briefly describe what is the dust module in CMAQ.

**We provided a brief description of the CMAQ in-line windblown dust module and a citation for it in section 3.1 in the revised manuscript. Please see our response for question 15.**

26. Figure 7, caption: ": : :alternative emission estimated according to the GOCIderived AOD."

**As the reviewer suggested, we corrected the caption for Figure 8 in the revised manuscript.**